# Evolutionary highways to persistent bacterial infection

Jennifer A. Bartell [1], Lea M. Sommer [2], Janus A.J. Haagensen[1], Anne Loch[1], Rocio Espinosa[1], Søren Molin[1] & Helle Krogh Johansen [2,3]

Persistent infections require bacteria to evolve from their naïve colonization state by optimizing fitness in the host via simultaneous adaptation of multiple traits, which can obscure evolutionary trends and complicate infection management. Accordingly, here we screen 8 infection-relevant phenotypes of 443 longitudinal *Pseudomonas aeruginosa* isolates from 39 young cystic fibrosis patients over 10 years. Using statistical modeling, we map evolutionary trajectories and identify trait correlations accounting for patient-specific influences. By integrating previous genetic analyses of 474 isolates, we provide a window into early adaptation to the host, finding: (1) a 2–3 year timeline of rapid adaptation after colonization, (2) variant "naïve" and "adapted" states reflecting discordance between phenotypic and genetic adaptation, (3) adaptive trajectories leading to persistent infection via three distinct evolutionary modes, and (4) new associations between phenotypes and pathoadaptive mutations. Ultimately, we effectively deconvolute complex trait adaptation, offering a framework for evolutionary studies and precision medicine in clinical microbiology.

[1] The Novo Nordisk Foundation Center for Biosustainability, Technical University of Denmark, 2800 Kgs. Lyngby, Denmark. [2] Department of Clinical Microbiology, Rigshospitalet, 2100 Copenhagen Ø, Denmark. [3] Department of Clinical Medicine, Faculty of Health and Medical Sciences, University of Copenhagen, 2200 Copenhagen N, Denmark. These authors contributed equally: Jennifer A. Bartell, Lea M. Sommer. Correspondence and requests for materials should be addressed to J.A.B. (email: jenbar@biosustain.dtu.dk) or to L.M.S. (email: lemad@biosustain.dtu.dk)

Bacteria have spent millennia evolving complex and resilient modes of adaptation to new environments, and some species effectively deploy these skills as pathogens during colonization within human hosts[1–3]. Due to gradual increases in fitness via accumulating genetic and epigenetic changes, it has been difficult to pinpoint overarching drivers of adaptation (from systems-level traits down to individual mutations) that reliably signal fitness[4]. Distinct populations may travel along the same predictable path to successful long-term persistence within a host, but other unique sequences of multi-trait adaptation can be equally optimal[5] in a complex, fluctuating environment[6]. This is even more relevant in a clinical context where dynamic selection pressures are applied via therapeutic treatment intended to eradicate infection.

Even for a well-studied model system of bacterial persistence leading to chronic infection such as the airway infections of cystic fibrosis (CF) patients, evolutionary trajectories remain difficult to map due in part to competing modes of evolution. We know from laboratory evolution studies in highly controlled conditions that these multiple modes are at work and induce substantial phenotypic adaptation to minimal media within the initial 5,000–10,000 generations[4,7,8], but only an estimate is available of the timeline of adaptation in the complex CF lung environment[9]. Multiple recent studies have shown a high degree of population heterogeneity in chronic CF infections that could be influenced by competing evolutionary modes, but past consensus has been that select traits converge towards similar evolved states during most CF infections (e.g., loss of virulence and increase in antibiotic resistance)[3,10–12]. This convergence can be complex and drug-driven, as recent studies have shown development of collateral sensitivity to antibiotics (treatment with one drug can induce reciprocal changes in sensitivity to other drugs);[13] this illustrates that a single selection pressure can reversibly affect multiple other traits, obscuring evolutionary trends. As in evolution in most complex, natural environments, persistent bacterial infections are influenced by strong and competing selective forces from very early in a patient's life. However, few studies have focused on the early periods of infection where environmental strains transition to successful pathogens in patient lungs.

Studies have assessed the genetic evolution of human pathogens in CF and identified specific genetic adaptations correlating with colonization and persistence[14–16]. However, only a few have linked genotypic and phenotypic changes[2,9,17,18], as this is especially challenging in natural populations. The genetic signature of adapting phenotypes is obscured over the course of evolution by the continuous accumulation of mutations and acclimatization by environment-based tuning of pathogen activity. Furthermore, it is inherently difficult to identify genotype-phenotype links for complex traits governed by multiple regulatory networks[19,20]. Consequently, we are far from the reliable prediction of phenotypic adaptation by mutations alone during evolution in a complex, dynamic environment such as airway infections in CF[19,21], and we propose that for now, phenotypic characterization is equally important. Analyses of infections of CF airways are an important platform for resolving these issues; approaches can be directly translated to other increasingly concerning persistent and chronic infections[1,18,22].

To address the complexity of pathogen adaptation in the host environment, we analyzed our phenotypic dataset using statistical methods that account for the environmental effects on patient-specific lineages (Generalized Additive Mixed Models—GAMMs) and assess adaptive paths traversing the evolutionary landscape from a multi-trait perspective (Archetype analysis—AA). We identify emergent patterns of bacterial phenotypic change across our patient cohort that depart from expected evolutionary paths and estimate the period of initial rapid adaptation during which the bacteria transition from a "naive" to an "evolved" phenotypic state. We further identify distinct and repeating trajectories of pathogen evolution, and by leveraging our prior genomics study of this isolate collection[16], we propose new associations between these phenotypic phenomena and genetic adaptation. We find that specific traits, such as growth rate and ciprofloxacin resistance, can serve as rough estimators of adaptation in our patients, while multi-trait modeling can map complex, patient-specific trajectories towards distinct evolutionary optima that enable persistence. Implementation of this trajectory modeling in general evolutionary studies might enable scientists to more easily define multi-trait evolutionary objectives and the genotype-phenotype relationships that enable their realization. Implementation as a diagnostic tool in patient care might enable clinicians to respond more quickly and effectively to evolving pathogens and inhibit the transition to a persistent infection.

## Results

**Evaluating pathogen adaptation in early stage infections.** The collection of 443 clinical *P. aeruginosa* isolates originates from a cohort of 39 youth with CF (median age at first *P. aeruginosa* isolate = 8.1 years) treated at the Copenhagen CF Centre at Rigshospitalet and captures the early period of adaptation, spanning 0.2–10.2 years of colonization by a total of 52 clone types. Of these isolates, 373 were previously characterized in a molecular study of adaptation[16]. Isolates were collected at the clinic from both the upper airways through sinus surgery samples, and the lower airways through expectorate (sputum) and bronchioalveolar lavage (BAL) samples. At study initiation, none of the patients were diagnosed as chronically infected by the CF center at Rigshospitalet, which is defined as elevated antibodies and/or multiple *P. aeruginosa* cultures spanning a period of six months[23,24]. Whenever a single clone type has been sampled over multiple sampling dates within a patient, we use the term "persisting colonization". The colonization time of an isolate is defined for each specific lineage, approximating the length of time since a given clone type began colonization of the CF airways in the specific patient. Importantly, our colonization time metric does not necessarily start at the true time zero, since a significant bacterial load is necessary for a positive culture. Our isolate collection also does not capture the complete population structure, but a previous study shows that 75% of our patients have a monoclonal infection persisting for years with mutations accumulating in a highly parsimonious fashion indicating unidirectional evolution[16]. Additionally, a metagenomic study of four patients from our cohort indicates that the single longitudinal isolates are representative of the major propagating subpopulation[25].

To obtain systems-level readouts of pathogen adaptation in the host and thereby assess multi-trait evolutionary trajectories, we present an infection-relevant characterization of our isolate collection entailing high-throughput measurements of 8 phenotypes: growth rates (in Luria-Bertani broth (LB) and artificial sputum medium (ASM)), antibiotic susceptibility (to ciprofloxacin and aztreonam), virulence factors (protease production and mucoidy), and adherence (adhesion and aggregation) (Figs. 1 and 2). We define adherence as a shared trend in adhesion and aggregation, which we associate with a biofilm-like lifestyle (see Methods for further discussion of limitations of these measures). These phenotypes are generally accepted to change over the course of colonization and infection of CF airways based primarily on studies of chronically-infected patients[10,17,26,27].

That is, an evolved isolate would grow slowly, adhere proficiently, be more likely to exhibit a mucoid and/or

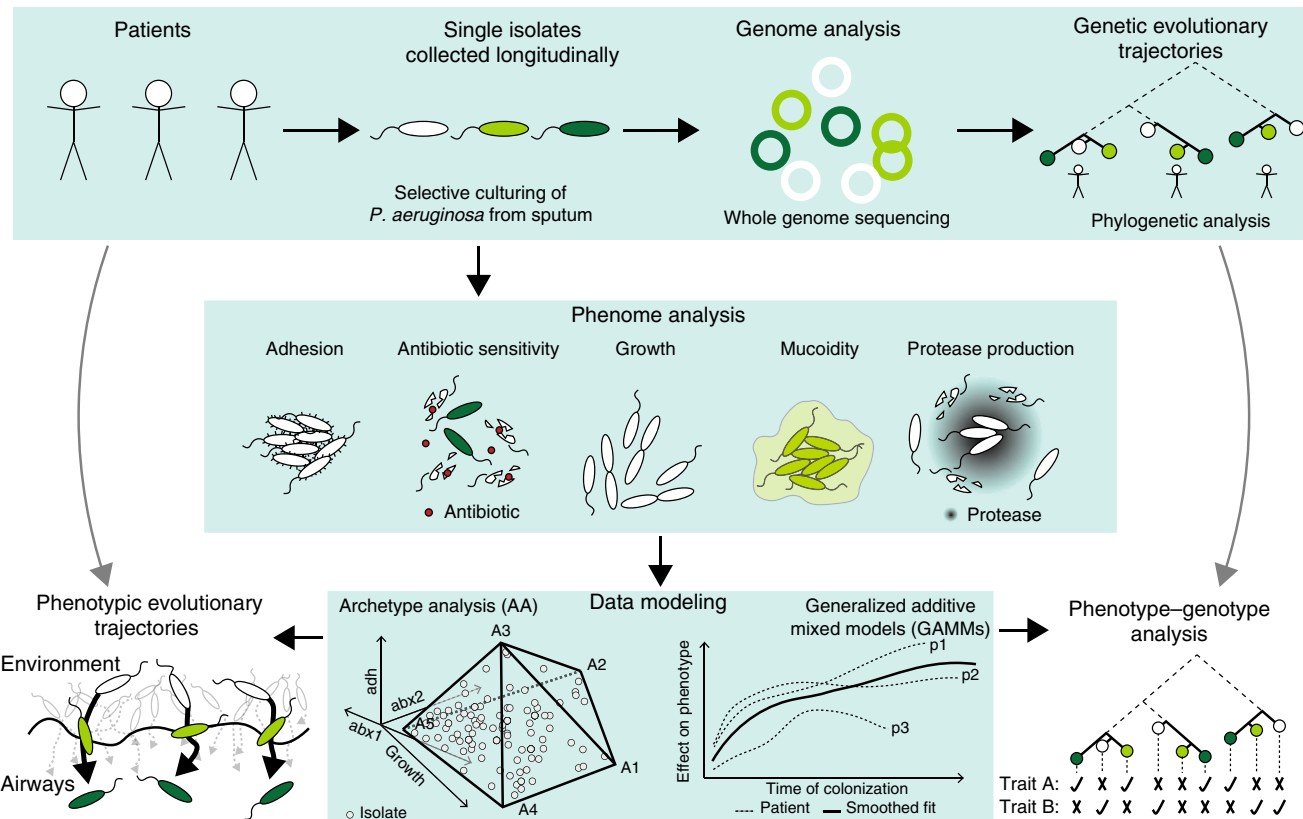

**Fig. 1** Study design. Upper panel: Every month, CF patients are seen at the CF clinic at Rigshospitalet in Copenhagen, Denmark. Here they deliver a sputum or endolaryngeal suction sample where selective microbiological culturing is performed[78]. The longitudinally collected isolates have been genome sequenced and analyzed previously[16]. Middle panel: Longitudinally collected isolates have been subjected to different phenotypic analyses for this study and are here (lower panel) analyzed using two data modelling approaches: Archetype analysis (AA) and Generalized Additive Mixed Model (GAMM). By integrating these approaches, we map dominant evolutionary trajectories and analyze mechanistic links between phenotypic and genetic adaptation

hypermutator phenotype, have reduced protease production, and resist antibiotics, in contrast to a naive isolate (Fig. 2b). However, simply ordering our measurements by colonization time does not illustrate an overarching adaptive trajectory from naive to evolved phenotypes (Fig. 2c). Instead, we see substantial heterogeneity, with isolates that resemble both naive and evolved phenotypic states throughout the study period. Given that we are investigating a unique collection from a young patient cohort that we track for a substantial period of colonization, this data fills the critical gap between studies of acute infections and chronic infections[28]. We are surprised to see naive phenotypes retained in late colonization as well as isolates in early colonization that deviate significantly from PAO1 phenotypes. However, a general pattern of heterogeneity is in alignment with previous studies of both *P. aeruginosa* and *Burkholderia* spp. infections[3,11,12]. To make sure that the presence of late naive isolates is not caused by a new infection with the same clone type, we investigated the molecular distance to the other isolates of the patient in question. Five lineages with isolates sampled later than three years after first detection of the clone type presented with seven or eight naive phenotypic characteristics out of the eight measured (DK13 from patient P0504, DK17 from P3804, DK41 from P7604, DK25 from P2605 and DK19 from P7204). In all cases, we found only a few SNPs separating late naive isolates from their nearest neighbors (0–2 SNPs[16]). However, in DK19 we found more SNPs separating two of the late naive isolates from the nearest neighbor than in the other cases. However, by looking at the maximum likelihood phylogeny including all identified SNPs of the DK19 clone type (present in multiple patients)[16] (Fig. 2a, rightmost panel), we are

convinced that this is not caused by a reinfection by the same clone type; these isolates are still more related to other isolates within the same patient than to isolates from other patients.

**A unique modeling approach.** Because our data is heterogeneous, we required specialized modeling approaches to account for specific environmental pressures and assess the boundaries of the evolutionary landscape. Previous studies have employed linear mixed models of phenotypic adaptation[29], and employed AA in the comparison of features of transcriptomic adaptation by *P. aeruginosa*[30]. Similar studies of multi-trait evolutionary trade-offs using polytope fitting have predicted the genetic polymorphism structure in a population[31]. We use related modeling methods to ensure that patient-specific effects are minimized, irregular sampling intervals are smoothed and a multi-trait perspective is prioritized by (1) modeling the dynamic landscape of multi-trait evolution using AA and (2) evaluating temporal correlations of phenotypic adaptation by fitting cross-patient trendlines using GAMMs (Fig. 1). We describe our approach below in brief, with more extended explanation available in both the Methods and Supplementary Notes 1 and 2.

With AA, we want to assess multi-trait adaptive paths within the context of the evolutionary landscape. We map these paths (or trajectories) by first fitting idealized extreme isolates (archetypes) located on the boundaries of the evolutionary landscape and then evaluating every other isolate according to its similarity to these idealized extremes. The archetypes are positioned at the corners of the principal convex hull (PCH), the

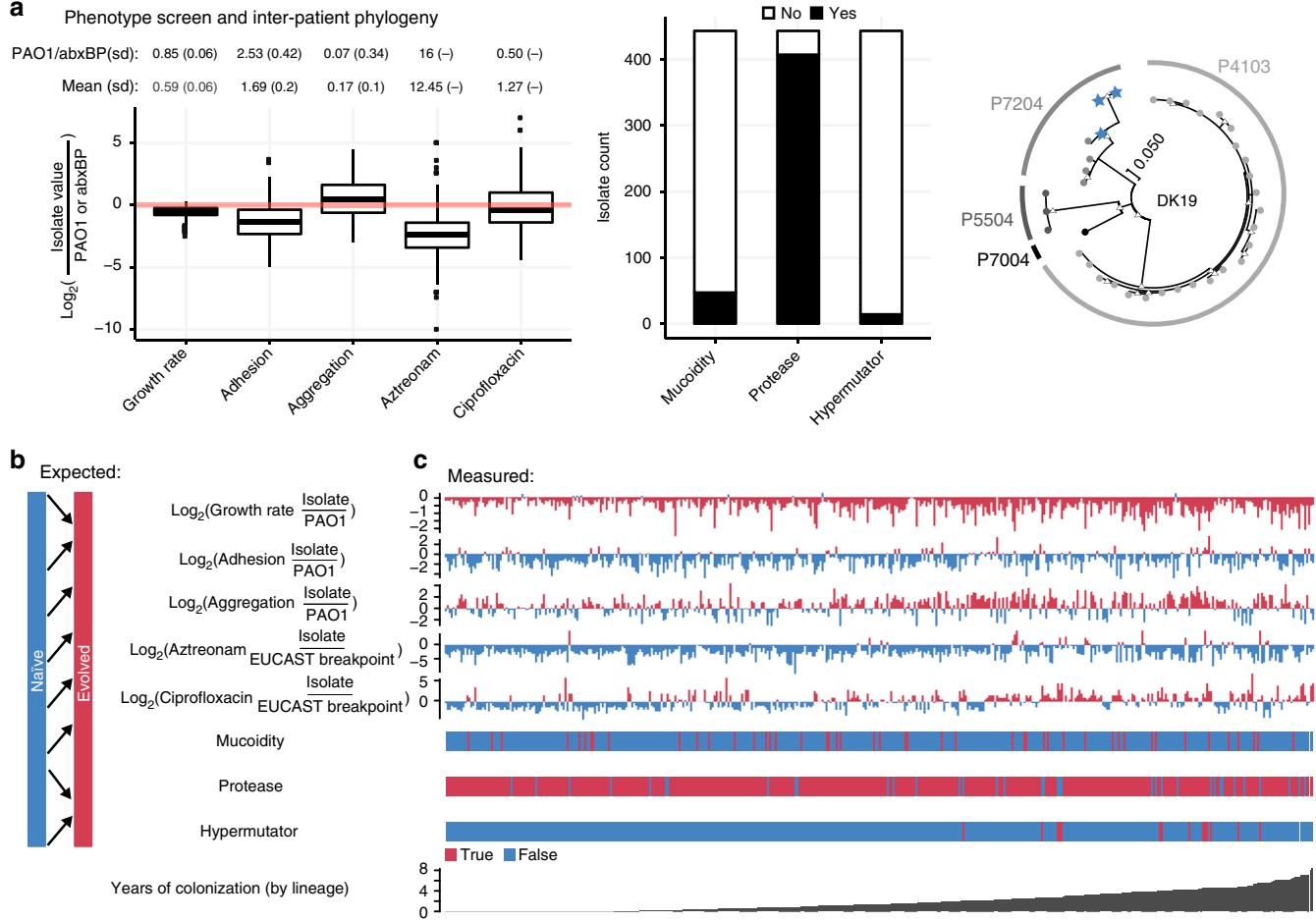

**Fig. 2** Phenotypic characterization. We present summary statistics of our phenotype screen including **a** mean and mean standard deviation for each phenotype over all isolates as well as the *P. aeruginosa* PAO1 value and antibiotic breakpoint we use for normalization, respectively (above boxplot). We also show boxplots of continuous normalized variables (including the median as the center line, first and third quartile box bounds and whiskers representing 1.5× inter-quartile range). We also show the overall count of isolates with presence/absence of mucoidity, protease and hypermutator phenotypes and a maximum likelihood phylogeny (1000 bootstraps) of the DK19 clone type; nodes marked with white triangles have bootstrap values >=50. Blue stars represent late (>3 years) phenotypically naive (7–8 naive phenotypes out of 8) isolates from patient P7204. Circles with different shades of grey represent isolates from patients marked on the outer edge of the circle. We then compare the **b** expected adaptation over time based on field consensus versus **c** the measured raw adaptation of our isolate collection over time. The X-axis represents the time since colonization of a specific lineage or "colonization time". Colors are linked with the expected change of the specific phenotype (**b**), so that blue denotes a "naive" phenotype and red denotes an "evolved" phenotype. For growth rate (in artificial sputum medium (ASM)), adhesion and aggregation, naive and evolved phenotypes are determined by comparison with the reference isolate PAO1 phenotype. For aztreonam and ciprofloxacin MIC, naive and evolved phenotypes are based on sensitivity or resistance as indicated by the EUCAST breakpoint values as of March 2017

polytope of minimal volume that effectively encapsulates our phenotype dataset[32] (Fig. 1, bottom panel). We conceptualize archetypes as the naive and evolved states of plausible adaptive trajectories and predict both the optimal number of archetypes and their distinct phenotypic profiles. We illustrate the AA by the 2D projection of our multi-trait model via a simplex plot, as shown in Fig. 3c[33].

With the GAMMs, we want to predict whether a given phenotype (the predicted variable) significantly correlates with other phenotypes or time (the explanatory variables). To do this we need to account for the effects of patient-specific environments and the effect of sampling time, while fitting trend lines for each trait (Fig. 1, bottom panel). This is done by fitting patient and time as random effects; we reduce the risk of overfitting by using a penalized regression spline approach with smoothing optimization via restricted maximum likelihood (REML)[34]. To avoid assumptions of cause-and-effect between our variables, we permute through different one-to-one models of all phenotypes,

and then reduce our models by combining only the statistically significant individual phenotypes into a multi-variable model. We further remove any phenotype that loses significance in the multi-variable model, assuming that it is correlated with a more impactful phenotype. From this point, all mentions of significance are obtained from the GAMM analyses with *p*-values < 0.01 based on Wald-type tests as described in[34,35], unless otherwise stated.

**Revealing multi-trait adaptation on a cross-patient scale**. AA predicted six distinctive archetypes sufficient to describe each isolate within the evolutionary landscape of 5 continuous traits as shown in Fig. 3a. We use only growth rate in ASM due to its correlation with growth rate in LB (Fig. 3d). The simplex plot of Fig. 3c highlights the standout features of each archetype by annotating according to the highest or lowest values for each phenotype across all archetype trait profiles (Fig. 3b). This simplex key illustrates that two archetypes resembled naive and un-evolved isolates with fast growth, antibiotic susceptibility, and low

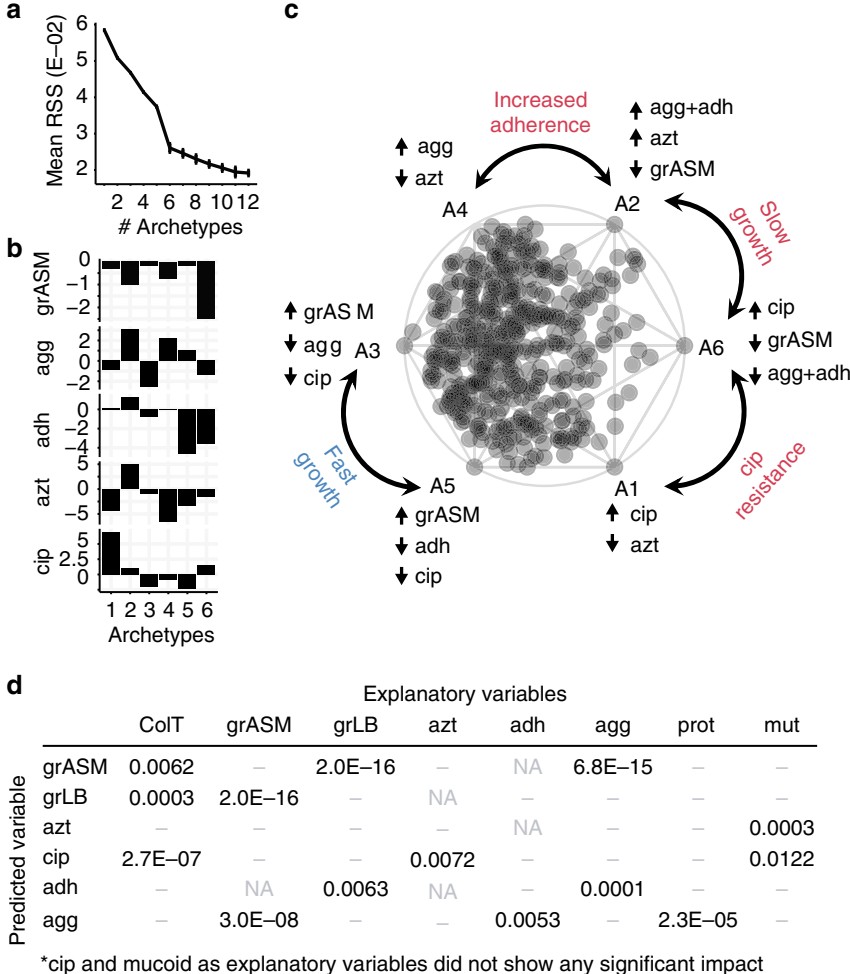

**Fig. 3** AA and GAMM models. We present a summary of the models underpinning our study of pathogen adaptation. **a** Screeplot showing the average residual sum of squares (RSS) for 25 iterations of each fit of a given number of archetypes. The "elbow" of the plot indicates that six archetypes are sufficient to model our dataset. **b** Characteristic trait profiles describing the five distinct phenotype levels that each of our 6 archetypes represents. We use the following abbreviations to represent our normalized data: grASM for growth rate in ASM, agg for aggregation, adh for adhesion, azt for aztreonam susceptibility, and cip for ciprofloxacin susceptibility. **c** Simplex plot of the AA showing the six archetypes (A1–A6) sorted by their characteristic growth rate (A3 and A5 vs A2 and A6), decreased sensitivity towards ciprofloxacin (A1 and A6), and increased aggregation and adhesion (A2 and A4). All further simplex visualizations are also sorted accordingly and can be interpreted using this key, which is annotated with the extreme phenotype values for each archetype. The complete analysis can be found in Supplementary Note 1. **d** $P$-values for GAMM models with multiple explanatory variables (columns) for the six predictor variables (rows) after model reduction. $P$-values are only shown for explanatory variables that showed a significant ($p$-value<0.01, GAMM with Wald-type tests) impact on the predictor in question. The complete analysis can be found in Supplementary Note 2

adherence (Archetype A3 and A5), while two others accounted for slow-growing evolved archetypes (A2 and A6), in accordance with the accepted paradigm[10,27]. A substantial portion of isolates in our study resemble the naive archetypes more closely than the evolved archetypes as indicated by their localization in the simplex plot (Fig. 3c, most isolates cluster on the left near the naive archetypes). This aligns with the infection stage of the patients included in this study. Importantly, we also find two regions in the simplex visualization which represent different focal points of adaptation: (1) an increase in adherence (A2 and A4) and (2) ciprofloxacin resistance (A1 and A6).

We also built a GAMM for each of our six continuous phenotypes to identify whether any of the other traits and time influenced it significantly across our patient cohort (Fig. 3d). First, we found that patient background had a significant impact on all predicted variables ($p < 0.01$, GAMM based on Wald-type tests), underlining the importance of accounting for the

environment and partially the genetic background of the lineages. When evaluating adaptation of the specific phenotypes, we found that the colonization time had a significant impact on both growth rate and sensitivity to ciprofloxacin but did not significantly influence sensitivity to aztreonam (Figs. 3c and 4a, b), which is a reflection of the current treatment regimen of the patients with regular administration of ciprofloxacin but not aztreonam[16,36].

**Phenotypic trends contrast with CF paradigms.** An important distinction between AA and GAMMs is that many isolates clearly cluster in AA according to phenotypes whose adaptation is not significantly influenced by time of colonization as shown by GAMMs. This contrast shows the importance of combining these approaches to understand our data. As an example, the biofilm-related metric of mucoidity does not significantly correlate with

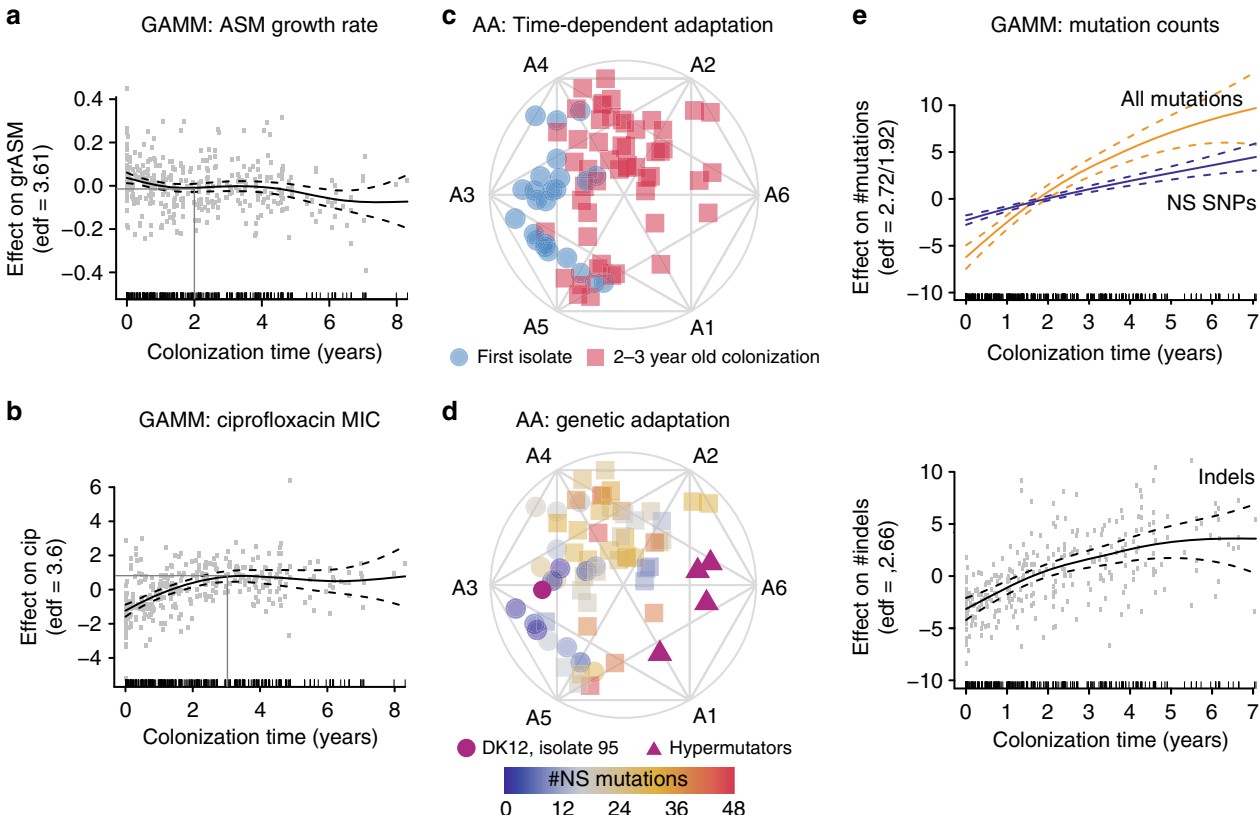

**Fig. 4** Rapid early adaptation. GAMMs illustrate the significant impact of the explanatory variable colonization time on **a** growth rate in ASM, **b** ciprofloxacin sensitivity in ASM, and **e** the accumulation of all mutations (orange), nonsynonymous SNPs (blue) and indels (insertions and deletions). We also illustrate the proposed initial adaptation period by dark grey reference lines in **a** and **b**. (**a/b/e** notation) GAMMs are illustrated by solid smoothed trendlines, dashed two standard error bounds, and gray points as residuals. Y-axes are labelled by the predictor variable on which the effect of colonization time of the clone type has been estimated as well as the estimated degrees of freedom (edf) (for the **e** upper panel the edf is ordered as all mutations/NS SNPs). Residuals have not been plotted in the upper panel of **e** for clarity reasons. X-axes are the colonization time in years and patients are included as random smooths together with time. A rug plot is also visible on the x-axis to indicate the density of observations over time. Simplex visualizations of AA show (**c**) naive trait alignment of the first isolate of the twenty patients where we have analyzed the first *P. aeruginosa* isolate ever cultured at the CF clinic (blue circles) in contrast to evolved isolates that have been cultured at year 2–3 of colonization (red squares, all patients of the dataset). We contrast this trait-based ordination with (**d**) genetic adaptation, shown by a color overlay of the number of nonsynonymous mutations present in each isolate. Isolate 95 (purple circle) of the DK12 clone type has a very high number of mutations (>100) because one isolate in that lineage (isolate 96) is very different from the remaining 11 isolates. For the GAMM analysis shown in Fig. 4e, we filtered out the mutations from the errant DK12 96 single isolate that affected the whole lineage. Hypermutators are marked by purple triangles

any other measured phenotype. Furthermore, neither adhesion nor aggregation correlates with colonization time for this population of young patients, though we see selection for adherence in a few specific patients via AA. That this is not a major trend in our data is surprising when we consider that a biofilm lifestyle is expected to be beneficial to persistence in chronically-infected patients[5,37–39]. Together, these results prompt further reassessment of common assumptions regarding the evolutionary objectives of *P. aeruginosa* in CF infections.

**Initial adaptation happens within 3 years of colonization.** We find that the routes to successful persistence and a transition to chronic infection are initiated early in infection[16,40]. The GAMMs indicate that a substantial change occurs in both growth rate and ciprofloxacin susceptibility during the first 2–3 years (5256–7884 bacterial generations[26]) of colonization as shown by the trendline slopes in this period (Fig. 4a, b). Using AA, we also see a substantial shift from naive towards evolved archetypes as shown by the broad distribution of isolates reaching the outer simplex boundaries by year 3 (Fig. 4c), further confirming the rapid adaptation shown by the GAMMs. While the first isolate of

each patient in our collection may not represent the true start of adaptation given sampling limitations, the window of rapid adaptation is still likely substantially contracted compared to the previous estimate of within 42,000 generations[9].

Interestingly, the four hypermutator isolates arising in the early adaptation window do not alone define the AA boundary, indicating that the acquisition of a high number of mutations does not explain all extreme phenotypes (Fig. 4d, full dataset in Supplementary Figure 1). To further evaluate parallels between phenotypic and genetic adaptation, we investigated the accumulation of nonsynonymous mutations in coordination with archetypal relationships (Fig. 4d, e). We used the isolates representing the first *P. aeruginosa* culture from a patient as the reference point for identification of accumulating mutations. We observed that most of the first isolates with 0–30 mutations aligned with naive archetypes, and 2–3-year-old isolates with 9–48 mutations extended to the outer boundaries of adaptation (A2, A6, and A1) (Fig. 4c, d). We also observed the persistence of WT-like genotypes with few mutations alongside evolved genotypes (Fig. 4d). Thus, we find discordant molecular and phenotypic adaptation from a multi-trait perspective.

The dN/dS ratio of all lineages with greater than 3 years colonization time ranged from 0.14 to 1.08 with an average of 0.54, indicating the dominance of negative selection (69% of lineages with a probability of neutral selection < 0.05, see Supplementary Data 2). When analyzing the entire dataset using GAMMs, we found a significant, near-linear relationship between colonization time and the number of nonsynonymous SNPs (Fig. 4e). However, accumulation of all nonsynonymous mutations appears logarithmic with accumulation slowing after 2 years; when we plot accumulation of indels alone, we see the likely driver of the logarithmic trend. When combined with the discordance found by AA, these findings support the theory that select beneficial mutations (for example, a highly impactful indel) can alone induce important phenotypic changes that improve fitness[41]. However, the likelihood of beneficial mutations presumably decreases over time as theorized previously[42] and other methods of adaptation also contribute, such as acclimation to the CF lung environment via gene expression changes[43,44].

**AA enables complex genotype-phenotype associations.** The obscuring of genotype-phenotype links via polygenic effects and the possible pleiotropic effects of single mutations is difficult to resolve, especially when working with complex traits. However, using our multi-dimensional perspective, we mapped a subset of 52 previously identified pathoadaptive genes—genes mutated more often than expected from genetic drift and thus assumed to confer an adaptive advantage during infection[16,45]. By overlaying nonsynonymous mutations on AA simplex plots, we evaluated the impact of mutation of the following pathoadaptive genes: (1) *mexZ* (the most frequently mutated gene) and other repressors of drug efflux pumps (*nfxB* and *nalD*), (2) mucoidity regulators *mucA* and *algU* and the hypothesized infection-state switching *retS/gacAS/rsmA* regulatory pathway previously examined from a genetic adaptation perspective[16,46], and (3) ciprofloxacin resistance genes *gyrA* and *gyrB*[47–49]. Isolates with *mexZ* mutations are broadly distributed by AA, so we analyzed *mexZ* mutants in combination with other pump repressor gene mutations. Even double-mutant isolates (grouped by efflux pump associations) showed diverse phenotypes via AA, though we noted a unique distribution of the many isolates impacted by a mutation in *nfxB* (Supplementary Figure 3, Fig. 5b). We saw no obvious spatial correlations with mutations linked to mucoidity regulation via AA (Supplementary Figure 2), paralleling mucoidity's lack of significance in our GAMM analyses. However, the isolate distributions of *retS/gacAS/rsmA* and *gyrA/B* mutants were striking in their spatial segregation (Fig. 5a, b).

**Differential evolutionary potential via resistance mechanisms.** The primary drivers of ciprofloxacin resistance in *P. aeruginosa* are theorized to be mutations in drug efflux pump repressor *nfxB* and the gyrase subunits *gyrA* and *gyrB* of the DNA replication system[47–49]. We would therefore expect isolates with mutations in these genes to cluster around archetypes A1 and A6 characterized by high ciprofloxacin minimal inhibitory concentrations (MICs) (Fig. 3c). However, AA illustrates a broad distribution of *gyrA/B* mutants among archetypes, and a contrasting narrow distribution of *nfxB* mutants (Fig. 5a, b, left panel). In association, we see a range of ciprofloxacin resistance levels associated with affected isolates both across and within patient lineages, and no dominant mutations/mutated regions repeating across lineages (Fig. 5a, b, right panel). The incidence of resistance was equal at 78% of affected isolates (54 out of 69 resistant gyrase mutants vs. 37 out of 47 resistant *nfxB* mutants based on the European Committee on Antimicrobial Susceptibility Testing

(EUCAST) breakpoint). However, the persistence of these respective mutations in affected lineages was dissimilar. Generally, *nfxB* mutation occurred earlier in lineage evolution and persisted in fewer lineages compared to *gyrA/B* mutations. This likely contributes to *nfxB*'s distinctive band-like distribution via AA which suggests an evolutionary restriction associated with sustaining the mutation.

Interestingly, we noted that isolates with a *gyrB* mutation (22 isolates alone or 14 in concert with *gyrA* mutation) are concentrated closer to biofilm-linked archetypes A2 and A4 than isolates with only a *gyrA* mutation (33 isolates). To our knowledge, there is no direct relationship between *gyrB* and the capability to adhere[49]. This positive association of *gyrB* on adhesion was confirmed by GAMM, but when we moved the two SNPs affecting the most isolates in both *gyrA* and *gyrB* (2 lineages each, Supplementary Figure 4) into lab strain *P. aeruginosa* PAO1, we did not find the same association (Supplementary Figure 5) ($p$-values > 0.05, ANOVA with Tukey correction, $F$ $(4,10) = 0.233$). Furthermore, one *gyrA* (G259A MIC: 0.5) and one *gyrB* (G1405T MIC: 0.5) mutant strain showed resistance towards ciprofloxacin, while PAO1 (MIC: 0.064) and the others did not (*gyrA*-C248T MIC: 0.25 and *gyrB*-C1397T MIC: 0.19), compared to the EUCAST breakpoint of 0.5. We then looked for co-occurring mutations in biofilm-linked genes in the *gyrB*-mutated lineages; for all but one lineage, there was no obvious explanation for increased adhesion. Ultimately, this association underlines the impact that genetic background and the multi-genetic signature of biofilm regulation can have on the identification of links between genotype and phenotype[50].

**Infection trajectory reversal via a regulatory switch.** The functional model of the *retS/gacA/gacS/rsmA* regulatory system is theorized to be a bimodal switch between acute and chronic infection phenotypes[46,51]. Posttranscriptional regulator *rsmA* activates an acute infection phenotype characterized by planktonic growth and inhibits a non-motile biofilm lifestyle. *retS* mutants are preserved in many lineages because they repress *rsmA* via the *gacA/S* two-component system, promoting a chronic infection phenotype. However, our previous genetic analysis[16] unexpectedly showed that multiple evolving lineages gained a subsequent mutation in *gacA/S* that often appeared years after the *retS* mutation. Despite the complexity of this regulatory system, we show a clear phenotypic separation between clinical isolates that are *retS* mutants versus *retS* + *gacA/S* mutants via our AA model (Fig. 5c, left panel). In this study, three of six patients with nonsynonymous mutations in this system have isolates which are *retS* + *gacA/S* double mutants (Fig. 5c, right panel). While *retS* mutants resemble the evolved archetypes (A1–2 and A6), all but one double-mutant cluster around the naive archetypes (A3–A5). According to patient-specific trajectories, this reversion happens after an initial migration towards evolved archetypes. Because of the limited isolates and patients affected, we did not follow up with additional GAMM analyses of the effect of these mutations on different phenotypes.

This unexpected phenotypic reversion to an acute infection state does not easily reconcile with theories about persistence via convergence towards a chronic phenotype. However, over time some patients are colonized by new clone types and/or other pathogens; this could require re-establishment of a colonization mid-infection and thus induce the population to revert towards an acute infection state where fast growth and motility improve its ability to compete.

**Infections persist via distinct routes of adaptation.** Given the above insights from lineage-based analysis, we further

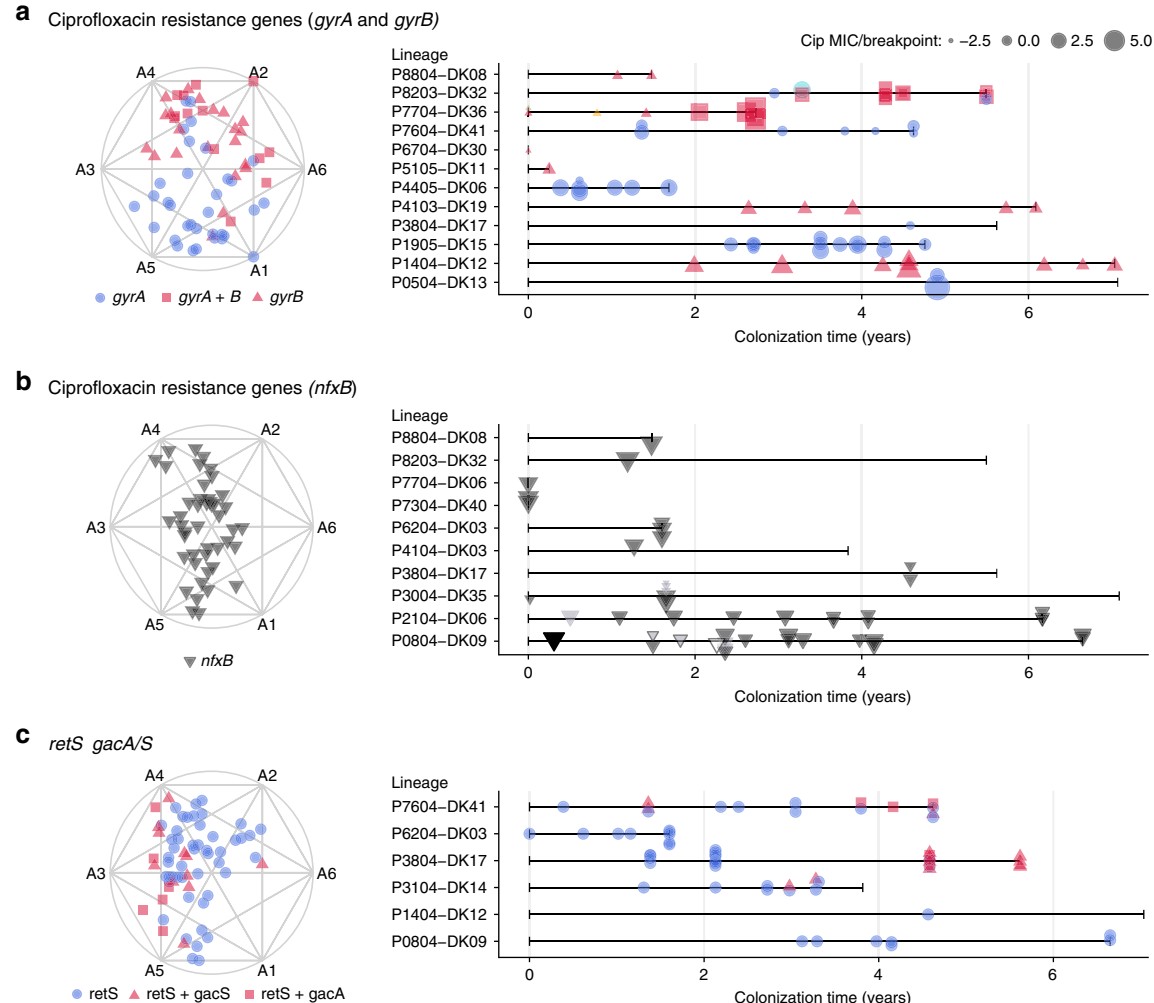

**Fig. 5** Mechanistic links, *gyrA/gyrB/nfxB* and *retS/gacAS/rsmA*. We use AA to illustrate phenotypic separation by isolates affected by distinct mutations in ciprofloxacin resistance genes *gyrA*, *gyrB*, and *nfxB* and the *retS/gacAS/rsmA* regulatory system. (**a**, **b**, left panel) As visualized by AA simplex plots, the diversity of trait profiles associated with isolates with mutations in DNA gyrase (*gyrA/B*) is in stark contrast to the constrained band of *nfxB*-mutated isolates. Mutations in DNA gyrase and *nfxB* do not co-occur in the same isolate but co-occur in different isolates of two lineages (patient P8804, genotype DK08 and patient P8203, genotype DK32). The differences in time of appearance during the colonization period and persistence of *gyrA/B* mutant isolates versus *nfxB* mutant isolates is shown in the lineage timelines plotted in the right column for *gyrA/B* (**a**, right panel) versus *nfxB* (**b**, right panel). Furthermore, *gyrB*-mutated isolates cluster more closely with A2 and A4 than *gyrA* mutated isolates, indicating a potential association with adhesion; GAMM predicts that *gyrB* mutation has a significant impact on adhesion (*p*-value « 0.01, GAMM with Wald-type tests). (**c**, left panel) Mutations in the *retS/gacAS/rsmA* system show a clear phenotypic change when *retS* is mutated alone (blue circles) or in combination with *gacA* or *gacS* (red squares and circles). The associated lineage plot (**c**, right panel) shows the appearance of double mutations (*retS* + *gacA/S*) after a colonization period by *retS* mutated isolates in three patient lineages. (**a/b/c** – lineage plot notation) Multiple isolates may be collected at the same sampling date based on differences in colony morphology or collected from different sinuses at sinus surgery, which explains the vertical overlap of isolates for some lineages. Lineage length is based on the span of time for which we have collected isolates and is indicated by gray bracketed lines, with only isolates affected by a mutation of the gene of interest plotted using shape and color (see legend of simplex plots). If multiple unique mutations occur in a lineage, this is specified by differential shading. (**a/b** only) Symbol size indicates the level of resistance to ciprofloxacin

investigate lineage influences by mapping patient-specific adaptive trajectories. We find three overarching modes of evolution that *P. aeruginosa* can utilize to persist successfully in individual patients: (1) convergent evolution, (2) directed diversity or (3) general diversity. Figure 6a, d shows examples of adapting lineages employing these modes. We see rapid convergent evolution towards a single archetypal endpoint (A3/A5 towards A1) of ciprofloxacin resistance in patient P5304 (Fig. 6a). Diverse isolates appear to move across the same general plane of multiple archetypal endpoints (A3/A4 towards A2/A6) towards increased adhesion and aggregation in patient P4104 (Fig. 6b), which we term "directed diversity". No directionality is apparent in the

diverse isolates of the trajectory of patient P0804 (Fig. 6c), which we term "general diversity". In the complex trajectory of patient P1404 (Fig. 6d), the genotypic distinction of the young isolate near A4 indicates that the persisting sublineage initiates with the isolate near A3, after which it gains a *gyrB* mutation guiding the trajectory towards ciprofloxacin resistant A1. This mutation is retained during the subsequent shift towards A2, characterized by increased adherence and decreased sensitivity to aztreonam. These results illustrate the diverse adaptive trajectories followed by *P. aeruginosa* in our patient cohort, which connect distinct start and endpoints of adaptation yet enable equally lengthy periods of persistent colonization.

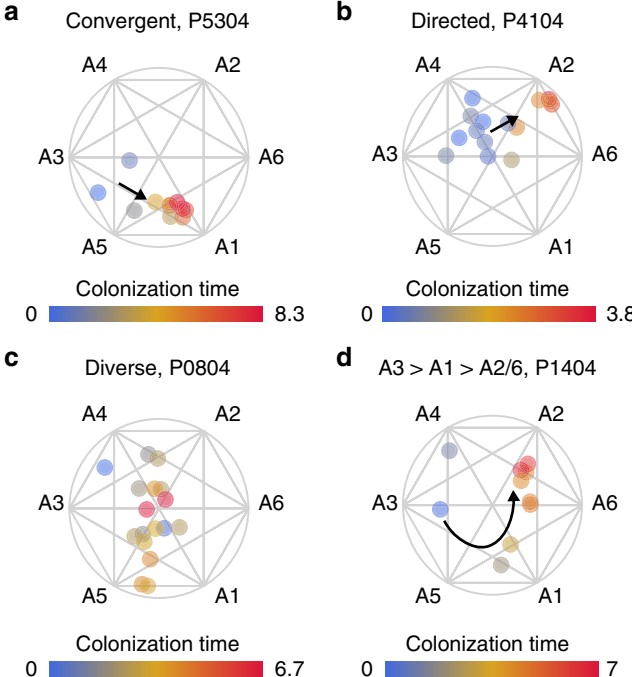

**Fig. 6** Evolutionary trajectories guided by different adaptation objectives. We present four different trajectories showing modes of evolution found in multiple patients: **a** Convergent evolution driven primarily by changes of a single phenotypic trait (decreased ciprofloxacin sensitivity). **b** Directed diversity with early/naive isolates showing a population moving in a broad and diverse plane from naive archetypes towards evolved archetypes. **c** General diversity where the population has no clear evolutionary trajectory. **d** A special case of convergent evolution with one outlier isolate (isolate 96 of DK12) but an otherwise clear trajectory first towards ciprofloxacin resistance and afterwards a gain in adhesive capabilities

Here, we draw specific examples from patients with high sampling resolution and at least 3 years of infection within our cohort, but to capture the full spectrum of evolutionary trajectories and the incidence of different evolutionary modes will require more uniform cross-cohort sampling that also addresses population dynamics as well as the inclusion of more patients. With these expansions, we theorize that distinctive evolutionary trajectories will correlate with infection persistence and patient outcomes.

## Discussion

Complex mutation patterns are an inherent byproduct of evolution driven by both genetic drift and selection, and result in equally complex adaptive trajectories that lead to persistence. Phenotype represents the cumulative systems-level impacts of these mutation patterns. We therefore emphasize the value of classical phenotype-based investigations as a highly relevant complement to genomics approaches. By integrating these perspectives via our statistical modeling framework, it is possible to identify consistent pan-cohort trends while illuminating complex patient-specific patterns and their genetic drivers. This approach could also be valuable in assessing evolution-based scenarios such as interpretation of laboratory evolution experiments, investigations of long-term microbiome fluctuations and studies of evolving clonal populations in other natural environments.

We identify signatures of adaptation that, when compared with prior studies, contrast current paradigms of beneficial adaptive phenotypes[10,17,26,27]. We achieve this via our approach

of tractable high-throughput in vitro assays despite our inability to replicate in vivo conditions (especially at this scale). As an example, mucoidity is used as an important biomarker of chronic infection in the Copenhagen CF Centre[52]. However, in this study, we do not identify any correlation with other measured phenotypes. We hypothesize that the rate of adaptation and relative benefit of this phenotype may vary significantly and be sensitive to temporal stresses such as antibiotic treatment. In support of our findings, others have recently shown that the longitudinal relationship between mucoidity and a clinical diagnosis of chronic infection is not as direct as previously expected[53]. We do see an expected decrease in growth rate and development of ciprofloxacin resistance, but it is characterized by an initial period of surprisingly rapid adaptation (approximately 5256–7884 bacterial generations[26]) contrasting with prior estimates for CF adaptation (~42,000 generations[9]). In fact, our observations align more with those from laboratory evolution experiments such as Lenski's long-term *Escherichia coli* evolution, where an initial phase of rapid fitness improvement was measured before 10,000 generations[7,8]. Our timeline of *P. aeruginosa* adaptation to the CF airways provides a valuable estimate of a narrow treatment window based on analysis at high temporal resolution, which aligns with the early aggressive antibiotic therapy used in the Rigshospitalet CF Centre[36].

While specific traits show cross-patient convergence (growth rate and ciprofloxacin resistance), we highlight remarkable diversity both within and across patients. In addition to convergent and directed evolution, we thus emphasize the maintenance of general diversity as a useful evolutionary mode of persistence as supported by prior observations of resilience in diverse populations[54–56]. The fact that we are only working with single isolates means that our findings may be partly obscured by subpopulation diversity and we are likely only observing the major changes as also suggested in Sommer et al. (2016)[25]. Due to the structure of our data, which includes both transient lineages and persisting lineages and irregular sampling constrained by patient infection state, we have likely undersampled the bacterial population in many patients. With further sampling, some patients currently grouped within the 'diverse' evolutionary mode could ultimately exhibit more directed evolution. However, we expect general diversity to be preserved in some patients as there is prior evidence of both diversity[11,12] and convergence[13] of phenotypes in CF infections. Among our patient-specific trajectories, we also find varying routes within these categories of evolution that enable the successful persistence of different patient lineages. We provide a quantitative approach to monitoring infection state via patient-specific trajectories, which can offer important insights into bacterial response to treatment. We aim to eventually class patients within a broader archetype model structured according to the extreme phenotypes and archetypal states of chronic isolates, which will allow relative grading of infection progression in visualizations accessible to clinicians. Incorporating records of patient treatment and response to our assessment of adaptive trajectories may further advance precision medicine in clinical microbiology.

Our study underlines the necessity of a multi-trait perspective, as individual mutations may have pleiotropic effects and obscure genetic signatures while accumulating over time[19]. Our genotype-phenotype associations support the theory that specific mutations confer unique evolutionary restrictions to adaptive trajectories; these restrictions impact the fixation of other mutations or adaptation of other traits, but genetic background and host-specific evolutionary pressures influence the type and degree of restriction[4]. By mapping phenotypic trajectories, we can identify both underpinning genetic changes and complex trait adaptations that signal the impact of selection pressures on individual infections.

## Methods

**The isolate collection**. The current isolate library is comprised of 443 longitudinally collected single *P. aeruginosa* isolates distributed within 52 clone types collected from 39 young CF patients treated at the Copenhagen CF Centre at Rigshospitalet (median age at first *P. aeruginosa* isolate = 8.1 years, range = 1.4–24.1 years, median coverage of colonization: 4.6 years, range: 0.2–10.2 years). This collection is a complement to and phenotypic extension of the collection previously published[16] and captures the period of initial rapid adaptation[7–9], with 389 isolates of the previously published collection included here in addition to 54 new isolates. To build a homogeneous collection for our study of evolution, we excluded two patients with a sustained multi-clonal infection. For the GAMM analysis, we excluded isolates belonging to clone types present in a patient at two or fewer time-points, unless the two time-points were sampled more than 6 months apart. The isolates not included in the previous study have been clone typed as a routine step at the Department of Clinical Microbiology at Rigshospitalet in order to confirm lineage association (variants have not been called). This clone type identification was performed in short by aligning de novo assemblies and demarcating clone types by >10,000 differential SNPs[16], and the sequencing was carried out as follows: DNA was purified from over-night liquid cultures of single colonies using the DNEasy Blood and Tissue Kit (Qiagen), libraries were made with Nextera XT and sequenced on an Illumina MiSeq using the v2 250 × 2 kit. The dN/dS ratio was calculated assuming that a nonsynonymous change is three times more likely than a synonymous change. The probability of neutral selection was calculated based on a cumulative binomial distribution probability. All variants assessed in this analysis were previously called and published in Marvig et al.[16].

**Ethics approval and consent to participate**. The local ethics committee at the Capital Region of Denmark (Region Hovedstaden) approved the use of the stored *P. aeruginosa* isolates: registration number H-4-2015-FSP. All patients have consented to study of their bacterial samples under the supervision of the Copenhagen CF Center. For patients below 18 years of age, informed consent was obtained from their parents. The study was carried out in accordance with the approved guidelines and the University Hospital Rigshospitalet approved the experimental protocol.

**Phenotypic characterizations**. For all phenotypes except the antibiotic MIC tests, phenotypic analysis was carried out by replicating from a 96 well plate pre-frozen with overnight cultures diluted with 50% glycerol at a ratio of 1:1 and four technical replicates were produced for each isolate. We acknowledge that hypermutators are not a true in vitro measured phenotype, as this characteristic is defined only by the non-synonymous mutation of either *mutS* or *mutL*, both initiator proteins of the mismatch repair system of *P. aeruginosa*, which is linked to an increased number of mutations in our collection[16].

**Phenotypic characterizations—growth rate**. Isolates were re-grown from frozen in 96 well plates in 150 μl media (LB or ASM[57]) and incubated for 20 hours at 37 °C with $OD_{630nm}$ measurements every 20 minutes on an ELISA reader. Microtiter plates were constantly shaken at 150 rpm. LB growth rates were first assessed by manual fitting of a line to the exponential phase of the growth curve. This dataset was then used to confirm the accuracy of R code that calculated the fastest growth rate from each growth curve using a sliding window approach where a line was fit to a 3–9 timepoint interval based on the level of noise in the entire curve (higher levels of noise triggered a larger window to smooth the fit). To develop an automated method of analyzing the ASM growth curves, which are much more noisy and irregular than the LB growth curves across the collection, we used standardized metrics for identifying problematic curves that we then also evaluated visually. Curves with a maximum OD increase of <0.05 were discarded as non-growing. Curves with linear fits with an $R^2$ of <0.7 were discarded as non-analyzable, and a small number of outlier curves (defined as curves analyzed for growth rates of 1.5 times the mean strain growth rate) were also discarded. Examples of our analyzed curves are shown in Supplementary Figure 6 and all visualizations are available upon request.

**Phenotypic characterizations—adherence measures**. The ability to form biofilm is a complex trait that is impacted by multiple factors, such as the production of polysaccharides, motility and the ability to adhere[58–60]. In this study, we have measured adhesion to peg-lids and estimated the ability to make aggregates. Both traits have been linked with an isolate's ability to make biofilm[61,62]. Because of this, we are using these two measures as an estimate of our isolates' ability to make biofilm. However, because we are aware of the complexity of the actual biofilm-forming phenotype, we have chosen to refer to this adhesion/aggregation phenotype as "adherence" and not biofilm formation.

Adhesion was estimated by measuring attachment to NUNC peg lids. Isolates were re-grown in 96 well plates with 150 μl medium where peg lids were used instead of the standard plate lids. The isolates were incubated for 20 h at 37 °C, after which $OD_{600nm}$ was measured and subsequently, the peg lids were washed in a washing microtiter plate with 180 μl PBS to remove non-adhering cells. The peg lids were then transferred to a microtiter plate containing 160 μl 0.01% crystal violet (CV) and left to stain for 15 minutes. The lids were then washed again three

times in three individual washing microtiter plates with 180 μl PBS to remove unbound crystal violet. To measure the adhesion, the peg lids were transferred to a microtiter plate containing 180 μl 99% ethanol, causing the adhering CV stained cells to detach from the peg lid. This final plate was used for measurements using an ELISA reader, measuring the CV density at $OD_{590nm}$. (Microtiter plates were bought at Fisher Scientific, NUNC Cat no. 167008, peg lids cat no. 445497).

Aggregation in each well was first screened by visual inspection of wells during growth assays in ASM and by evaluation of noise in the growth curves, resulting in a binary metric of aggregating versus non-aggregating. However, to incorporate this trait in our AA, we needed to develop a continuous metric of aggregation. Based on the above manual assessment, we developed a metric based on the average noise of each strain's growth curves. While we tested several different metrics based on curve variance, the metric that seemed to delineate isolates according to the binary aggregation measure most successfully was based on a sum of the amount of every decrease in OD that was followed by a recovery at the next time point (versus the expected increase in exponential phase and flatline in stationary phase). This value was normalized by the increase in OD across the whole growth curve, to ensure that significant, irregular swings stood out with respect to overall growth. This metric therefore specifically accounts for fluctuation—both a limited number of large fluctuations in $OD_{630nm}$ (often seen during stationary phase) as well as smaller but significant fluctuations across the entire curve (i.e., sustained irregular growth). While an imperfect assay of aggregation compared to available experimental methods[63], this high-throughput aggregation estimate showed a significant relationship with adhesion when analyzed with GAMMs (Fig. 3d), supporting its potential as a measure of adherence-linked behavior. We show examples of the measurement and comparison with binary aggregation data in Supplementary Figures 6–7.

**Phenotypic characterizations—protease production**. Protease activity was determined using 20 × 20 cm squared LB plates supplemented with 1.5% skim milk. From a master microtitre plate, cells were spotted onto the square plate using a 96 well replicator. Colonies were allowed to grow for 48 h at 37 °C before protease activity, showing as a clearing zone in the agar, was read as presence/absence.

**Phenotypic characterizations—mucoidity**. Mucoidity was determined using 20 × 20 cm squared LB plates supplemented with 25 μg/ml ampicillin. From a master microtitre plate, cells were spotted onto the square plate using a 96 well replicator. Colonies were allowed to grow for 48 h at 37 °C before microscopy of colony morphologies using a 1.25x air Leica objective. By this visual inspection, it was determined if a colony was mucoid or non-mucoid.

**Phenotypic characterizations—MIC determination**. MICs were determined for ciprofloxacin and aztreonam by E-tests where a suspension of each isolate (0.5 McFarland standard) was inoculated on 14 cm-diameter Mueller-Hinton agar plates (State Serum Institute, Hillerød, Denmark), where after MIC E-Test Strips were placed on the plate in accordance with the manufacturer's instructions (Liofilchem®, Italy). The antimicrobial concentrations of the E-tests were 0.016–256 μg/ml for aztreonam and 0.002–32 μg/ml for ciprofloxacin.

**Phylogenetic reconstruction**. Phylogenetic analysis was done using the program MEGA (v. 7.0.26)[64] to produce a maximum likelihood phylogeny with 1000 bootstraps of concatenated SNPs of the DK19 clone type as previously identified in Marvig et al. (2015)[16]. The tree presented in Fig. 2a is the tree with the highest log likelihood and bootstrap values >= 50 indicated by white triangles.

**Construction of *gyrA/B* mutants**. Four *P. aeruginosa* PAO1 mutants carrying point mutations in *gyrA* and *gyrB* were constructed: PAO1::*gyrA*^G259A, PAO1::*gyrA*^C248T, PAO1::*gyrB*^C1397T, and PAO1::*gyrB*^G1405T. A recombineering protocol optimized for *Pseudomonas* was adapted from Ricaurte et al. (2017)[65]. A PAO1 strain carrying a pSEVA658-ssr plasmid[66] expressing the recombinase *ssr* was grown to exponential phase with 250 rpm shaking at 37 °C. Bacteria were then induced with 3-methylbenzoate and electroporated with recombineering oligonucleotides. Cells were inoculated in 5 ml of glycerol-free Terrific Broth (TB) and allowed to recover overnight at 37 °C with shaking. Cip^R colonies were identified after streaking on a Cip-LB plate (0.25 mg L^−1) and sent for sequencing after colony PCR.

Each recombineering oligonucleotide contained 45 base pair homology regions flanking the nucleotide to be edited. Oligonucleotides were designed to bind to the lagging strand of the replichore of both genes and to introduce the mismatch in each mutation: G259A and C248T in *gyrA*, and C1937T and G1405T in *gyrB*, respectively. The recombineering primers used are the following:

Rec_gyrA_G259A -
G*C*ATGTAGCGCAGCGAGAACGGCTGCGCCATGCGCACGATGG
TGTtGTAGACCGCGGTGTCGCCGTGCGGGTGGTACTTACCGATCACG*T*C
Rec_gyrA_C248T -
A*G*CGAGAACGGCTGCGCCATGCGCACGATGGTGTCGTAGACCGCGa
TGTCGCCGTGCGGGTGGTACTTACCGATCACGTCGCCGACCAC*A*C
Rec_gyrB_C1397T -

C*C*GATGCCACAGCCCAGGGCGGTGATCAGCGTACCGACCTCCTGGa
AGGAGAGCATCTTGTCGAAGCGCGCCTTTTCGACGTTGAGGAT*C*T
Rec_gyrB_G1405T -
C*C*TCGCGGCCGATGCCACAGCCCAGGGCGGTGATCAGCGTACCGAa
CTCCTGGGAGGAGAGCATCTTGTCGAAGCGCGCGCCTTTTCGACG*T*T

**Modeling of phenotypic evolution.** To identify patterns of phenotypic adaptation while limiting necessary model assumptions that might bias our predictions, we chose to implement GAMMs, where the assumptions are that functions are additive and the components are smooth. These models allow us to account for patient-specific effects, thereby enabling us to identify trends in phenotypic adaptation across different genetic lineages and different host environments. Furthermore, to be able to simultaneously assess multiple phenotypes of each isolate from a systems perspective, we implemented AA, where each isolate is mapped according to its similarity to extremes, or archetypes, fitted on the boundaries of the multi-dimensional phenotypic space. This modeling approach allows us to predict the number and characteristics of these archetypes and furthermore identify distinctive evolutionary trajectories that emerge from longitudinal analysis of fitted isolates for each patient.

For all analyses, the time of infection is defined within each lineage as the time since the clone type of interest was first discovered in the patient in question. This is biased in the sense that the time since colonization can only be calculated from the first sequenced isolate of a patient. However, we have collected and sequenced the first isolate that has ever been cultured in the clinic for 20 out of the 39 patients.

Normalization of phenotypic values were carried out the following way for both AA and GAMM: ciprofloxacin and aztreonam MICs were normalized by dividing the raw MICs with the breakpoint values from EUCAST: ciprofloxacin breakpoint value: > 0.5 µg/ml, aztreonam breakpoint value: >16 µg/ml (EUCAST update 13 March 2017). This results in values above one equaling resistance and equal to or below one equaling sensitivity. The response and the explanatory variables were log2 transformed to get a better model fit for ciprofloxacin MIC, aztreonam MIC, Adhesion, and Aggregation. For the AA, Adhesion, Aggregation and growth rate in ASM was further normalized (before log2 transformation) by scaling the values by the values of the laboratory strain *P. aeruginosa* PAO1 such that zero was equivalent to the PAO1 phenotype measurement or the EUCAST MIC breakpoint. PAO1 was chosen to be the reference point of wild type phenotypes.

Because the mutations identified in our collection are based on our previous study[16] where mutations were called within the different clone types, we added a second filtering step to identify mutation accumulation within patients. The second filtering step removed mutations present in all isolates of a lineage (a clone type within a specific patient) from the analysis. Isolates without called variants (54 new isolates) were removed from all variant-reliant analyses.

All statistics were carried out in R[67] (version 3.4.0) using the packages *mgcv*[68,69] (version 1.8–18) for the GAMM analysis and *archetype*[33,70,71] (version 2.2) for the AA. Complementary packages used for analysis are: *tidyverse*[72] (version 1.1.1), *itsadug*[73] (version 2.3), *ggthemes*[74] (version 3.4.0), *knitr*[75] (version 1.16) and *kableExtra*[76] (version 0.6.1). We also referred to Thøgersen et al.[30] and Fernandez et al.[77] in the design of appropriate assessment methods for the final AA model. We include two R markdown documents that explain our modeling steps and further evaluation plots in detail (AA: Supplementary Note 1, GAMM: Supplementary Note 2), and summarize our methods below in brief.

**Data modeling—AA.** We evaluated several different model fitting approaches by varying the number and type of phenotypes modeled as well as the archetype number and fit method, using RSS-based screeplots of stepped fits of differing archetype numbers, explained sample variance (ESV), isolate distribution among archetypes, convex hull projections of paired phenotypes (all combinations), and parallel coordinate plots as metrics for choosing the best fit parameters and approach to accurately represent our data. Ultimately, we focused on 5 continuous phenotypes correlated with growth (growth rate in ASM), biofilm (adhesion and aggregation), and antibiotic resistance (aztreonam and ciprofloxacin MICs), which also were linked to relevant findings provided by the GAMM models. We used a root sum squared (RSS) versus archetype number screeplot of different fits to determine that a 6 archetype fit would produce the optimal model for this dataset.

We then performed 500 simulations of a 100 iteration fit using the robustArchetypes method[71], which reduces the impact of data outliers in fitting the convex hull of the data. We evaluated the mean ESV and the number of isolates with an ESV greater than 80% for the best model from each simulation in this study and differences in archetype characteristics to assess convergence, ultimately selecting the model with the second highest mean ESV (90.32%) and highest number of isolates with an ESV over 80% (87.13%); this model also closely resembled the other 10 top models of the simulation study. The order of archetypes around the simplex plot boundary obscures the true dimensionality of the isolate distribution by implying the archetypes are equidistant, so relationships between phenotypes are not always obvious. We re-ordered the archetypes in the simplex plot by growth rate and secondarily antibiotic resistance to improve clarity in the complex 6 archetype plot. This reordering was also justified when projecting the archetypes onto a PCA plot of the phenotypes (Supplementary Note 1). All simplex plots have also had the 11 isolates

with an ESV < 50% removed such that we are not drawing any conclusions from these poorly fit data (they are shown via simplex plot in the Supplementary Note 1).

**Data modeling—GAMMs.** For all phenotypes, GAMMs were used to identify evolutionary trends over time since first colonization. We correct for the patient environment and inconsistent sampling over time using a smooth random factor. Models were fitted in the following way: All continuously measured phenotypes included in the AA were fitted as a response variable (predicted or dependent variable in Fig. 3d) one-to-one, with both time as an explanatory or independent variable alone and combined with each of the phenotypes to account for potential time-dependence of the observations. Factorial/binary phenotypes were implemented as categorical functions and continuous phenotypes as smooth functions, allowing for non-parametric fits. Normally only one variable/phenotype of interest is used as the predictor while other alterable variables or factors are used as explanatory variables to explain or predict changes in the predictor. However, this requires a preconceived idea of a one-way-relationship where one variable (the predictor) is assumed to be affected by certain other variables (the explanatory variables), but where the explanatory variables cannot be affected by the predictor. By testing all phenotypes against each other, we avoid assumptions regarding the specific direction of relationships between the predictor variable and the explanatory variable. Furthermore, in using the GAMMs we prioritize accuracy of fitting but increase our risk of overfitting as a byproduct. We sought to counteract the risk of overfitting by the default penalization of fits inherent to the method used[68,69] and by model estimation via REML which has been found to be more robust against overfitting[34,69]. When significant relationships were identified in one-to-one models (*p*-value < 0.05, as based on Wald-type tests[34,35]), all significant explanatory variables were used to build a multi-trait model for the associated predictor. If select explanatory phenotypes were then identified as non-significant (*p*-value > 0.05) in the multi-trait model, they would be removed in a reduction step. To identify whether a reduced multi-trait model resulted in a better fit than the initial multi-trait model, a Chi-square test was carried out on the models using the compareML function of the R package *itsadug*[73] (Fig. 3d). The specific models and additional information can be found in Supplementary Note 2.

In demonstration of the utility of this approach, the multi-trait models of our five primary predictor phenotypes show that at least one explanatory phenotype has a statistically significant impact on the predictor phenotype. For all of the predictor phenotypes, multiple explanatory traits preserved significant impacts after model reduction steps (Fig. 3d and Supplementary Note 2). All mentions of significant relationships or correlations in the main text are obtained from the GAMM analyses with Wald-type test statistics presenting *p*-values < 0.01, unless otherwise stated. For information on deviance explained, $R^2$, and degrees of freedom for the individual models/variables, we refer to Supplementary Note 2.

**Reporting summary.** Further information on experimental design is available in the Nature Research Reporting Summary linked to this article.

**Code availability.** Data normalization, processing and construction of all models was performed in R (V. 3.4.0) as described above and all essential code for reproduction of these steps is provided in R Markdown format in Supplementary Note 1 (AA) and Supplementary Note 2 (GAMMs). Versions of the packages used are also indicated in the Supplementary Notes 1–2. These files also include code for replicating the model visualizations of Figs. 3a, d and 4a, c, e. Code to reproduce various secondary analysis figures is available on request.

## Data availability

The authors declare that all data and code necessary for supporting the findings of this study are enclosed in this paper. The entire dataset is enclosed in Supplementary Data 1, while the code used for the archetypal analysis (AA) and the generalized additive mixed models (GAMMs) are enclosed in Supplementary Note 1 (AA) and Supplementary Note 2 (GAMMs). Furthermore, we provide visualization and summary statistics of normalized data in Fig. 2. All genomic data is publicly available through the SRA database and has been published previously by Marvig et al.[16]. The accession numbers of all genomically analyzed isolates can also be found in Supplementary Data 1; the NCBI SRA study summary can be accessed at [https://trace.ncbi.nlm.nih.gov/Traces/sra/?study=ERP004853].

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

## Acknowledgements

H.K.J. was supported by The Novo Nordisk Foundation as a clinical research stipend (NNF12OC1015920), by Rigshospitalets Rammebevilling 2015–17 (R88-A3537), by Lundbeckfonden (R167-2013-15229), by Novo Nordisk Fonden (NNF15OC0017444), by RegionH Rammebevilling (R144-A5287) and H.K.J and L.M.S were supported by Independent Research Fund Denmark/Medical and Health Sciences (FTP-4183-00051). S.M. and J.A.B. were supported by the Novo Nordisk Foundation Center for Biosustainability (CfB), Technical University of Denmark. J.A.B was also funded by a post-doctoral fellowship from the Whitaker Foundation. We thank Katja Bloksted, Ulla Rydahl Johansen, Helle Nordbjerg Andersen, Sarah Buhr Bendixen, Camilla Thranow, Pia Poss, Bonnie Horsted Erichsen, Rakel Schiøtt, and Mette Pedersen for excellent technical assistance. We also thank Prof. Anders Stockmarr, Prof. Nina Jakobsen and Prof. Morten Mørup (DTU Compute) and Dr. Kevin D'Auria (Counsyl) for helpful discussions.

## Author contributions

S.M. and H.K.J. jointly supervised the study. J.A.H., S.M., and H.K.J. conceived and designed the experiments. J.A.H. performed all phenotypic screening with assistance from A.L.. R.E.P. performed the genetic engineering of isolate mutations. J.A.B. and L.M.S. conceived and performed all computational analysis and wrote the manuscript. J.A.H., S.M., and H.K.J. helped write the manuscript and provided revisions.

## Additional information

**Competing interests:** The authors declare no competing interests.

