## [Peer Review File · Nature Communications]

Reviewers' Comments:

Reviewer #1:

Remarks to the Author:

To study not only genetic adaptations to environmental changes but also phenotypic adaptation and to analyze how phenotype and genotype are linked to accomplish survival in host habitats is a huge challenge and promises to uncover deep and novel insight into bacterial strategies to survive during an infection process and to cause chronic infections. This study profits from a unique collection of fully genotyped *P. aeruginosa* clinical isolates that have been recovered from CF patients in a longitudinal study. These isolates were now subjected to extensive phenotyping as the starting point for the identification and characterization of complex trait adaptation strategies of *P. aeruginosa* to the lung of cystic fibrosis patients. For this purpose two models were used account for the identification of general adaptive paths that are followed by the clinical isolates and the effects of the environment on the patient-specific lineages. The impressive dataset and the original methods used in this study enabled the authors to address highly relevant questions. Their approach opens up new avenues to pursue evolutionary studies and to find new answers on mechanisms of persistence. This also promises the development of alternative strategies to fight chronic biofilm-associated infections in the future.

Specific comments:

I would have liked to see a color code in Fig 2C to distinguish early from late isolates.

Already in Fig 2 it becomes apparent that slow growth, aggregation and ciprofloxacin resistance are phenotypes that evolve in late isolates, whereas adhesion and azt resistance appear to not evolve during the course of an infection. How would the results of the AA model look like if those data were excluded?

The authors describe consistent pan-cohort trends. Since one can group the isolates that follow distinct group-specific convergent trends, are there genetic variations that can be assigned to those groups (e.g. by applying machine learning algorithms)?

I am not convinced by the explanation of the unexpected phenotype of a segregation of *retS* and *retS/gacA* mutants (line 339-344). The *retS/gacA* double mutation might drive the evolution of a phenotype that is not sufficiently described with the attributes slow growth, aggregation and resistance. It thus might be of particular interest to include virulence (cytotoxicity) data in the AA model in the future.

The "non-clustering" of the *gyrB* mutants (as opposed to the *gyrA* and *gyrA/gyrB* mutants) with the A1 and A6 archetypes might be explained by a lower resistance level due to the *gyrB* mutation. Did the authors test for the CIP MIC in their engineered PAO1 isolates?

Reviewer #2:

Remarks to the Author:

Pseudomonas aeruginosa causes chronic airway infections in an overwhelming majority of individuals with cystic fibrosis. Previous studies have identified a myriad of adaptations (both genetic and phenotypic) that occur during the transition from initial colonization to chronic or persistent infection. This study by Bartell and colleagues examines the evolutionary trajectory of *P. aeruginosa* during adaptation to the host environment by characterizing the phenotypic and genotypic changes that occur in longitudinal isolates from 39 patients spanning a 10-year period following initial colonization. This study is impressive in scope and provides an innovative analysis approach that is biologically informative and applicable to other complex and rapidly evolving systems. This study will be influential to both the CF and *P. aeruginosa* research fields as well as the larger fields of microbial ecology, pathogenesis and evolution. Major findings include evidence for 1) a period of rapid adaptation during the first 2-3 years of colonization, and 2) simultaneous directional and diversifying trajectories. Importantly, the authors demonstrate the power of studying both phenotypic and genetic adaptations

in order to fully understand the evolutionary trajectory of bacterial populations.

In general, this is a superb manuscript that could be further improved by addressing the following comments:

1. My primary criticism is that the major findings are difficult to identify due to the overly complex presentation style. At times the manuscript is redundant and often delves too deeply into peripheral or esoteric topics of evolution. The introduction and results sections should be significantly shortened to focus on the specific findings. Speculation and interpretations should be moved to a more formal discussion section.
2. The reliance on single isolates is problematic. While I appreciate that it is impossible to address this in a retrospective isolate collection, the authors should at least comment on how this could affect their interpretations.
3. The authors reference several previous studies that document diversification as an adaptive strategy. However, there is little mention of diversity in quorum sensing phenotypes, which has been well-studied in evolving CF *P. aeruginosa* populations.
4. Hypermutators are mentioned at several places in the results and supplement but they are never defined.
5. The authors state multiple times (lines 81, 151, 170) that they use a statistical approach that accounts for patient and environmental effects (this is also an example of the redundancy in the manuscript). Do linear models fail to give the same results? Is there evidence of patient and environmental effects? What do the authors anticipate the patient and environmental factors are?
6. The authors should comment on the complexity of using sputum as a proxy to study lung populations. CF lung disease is very heterogeneous with different regions of the infected lung having different selective pressures. Does sputum reflect all areas of the lung? Ref#12 is relevant to this point.
7. Figure 6 provides individual examples of different evolutionary trajectories. It is unclear what distinguishes the "Convergent" and "Directed" trajectories. Also, can the remaining patients be categorized based on these "overarching" modes of evolution? Is one mode more common than the others?

Reviewer #3:

Remarks to the Author:

The manuscript submitted to Nature Communications by Bartell et al. reports a detailed study investigating phenotypic changes in *P. aeruginosa* strains isolated from young CF patients over 10 years. One strength of the paper is the richness of the phenotypic and genotypic data. However, the main strength of the data is the elegant statistical modeling used to make the experimental findings more understandable and visible. I recommend this paper for publication after revision. Please find below my specific comments.

1. I had hard times assessing the quality of the experimental data used for the AA and GAMMs analysis. Figure 2A is confusing as it is. There are no error values for the reference strain. It is important to do a rigorous phenotyping/comparison by including the std values for the reference PAO1 strain. Some of the assays (such as adherence) are difficult and large error bars are expected but I would personally anticipate more precise MIC measurements. It was not clear to me whether the std values here corresponded the distribution across isolates or across biological replicates for a particular strain. Probably the former. Time axis of Figure 2B is a bit confusing as well.

2. I noticed an alternating pattern (naïve <-> evolved) in figure 2C particularly for Cipro resistance

and aggregation. Have the authors checked that?

3. There is no specific "persistence" definition. In the field of antibiotic resistance or cancer, persistence is used in a different context.

4. In several places, "early" phases of evolution and "rapid" evolution are used for the changes observed in the first few years. In population dynamics, a year for bacteria is quite long. Use of these terms should be justified by number of generations.

5. Growth rate measurements are done in ASM media but some other tests are done on MH agar. I don't know whether this can cause any discrepancy. I think there must be a consistency across different phenotype measurements (i.e. same media, same temperature etc). It will be helpful to show that (for a subset of isolates) phenotypic changes such as adherence do not significantly change in different media.

6. Authors observe retained naïve phenotypes in late colonization and justify this observation by referring to references #3,11, and 12. In my opinion, this is not the best way to address this. One very simple possibility is new infections which has to be ruled out by showing phylogenetic distances etc.

7. One interesting assumption in the analysis is assuming that the infections never completely clear. Is that true? Will the GAMMs analysis examining adaptation change if new infections from other CF patients are considered possible?

8. In couple of places, comparisons with Lenski papers are made but these are pretty vague. For instance, "In fact, our data resembles the rate of fitnesswithin the first 5000-10000 generations". This has to be elaborated. There are too many findings in Lenski papers and it is hard to remember them all and make the comparison.

9. AA analysis for phenotypic changes is really interesting but maybe (not surprisingly) it is not possible to use AA as effectively for the genetic changes. Looking at number of mutations is oversimplification.

10. Figure 4 is difficult to follow. The figure caption is too convoluted.

11. Figure 4A suggests a slight growth rate decrease over time. What is the slope if one simply does a linear fit? It is most likely zero. A linear fit can be included to train our eyes.

12. It would be nice to see the time periods where there is selection and there is only drift. Maybe a simple dN/dS analysis could help.

13. I couldn't come up with a better idea but figure 5 is organized in a very complicated way. Too many colors, shapes, and sizes.

14. Authors refer to distinct routes for adaptation. In my opinion, there are no distinct adaptation mechanisms. There is still selection and neutral drift as the two main drivers. Most likely, these different classes are special snapshots of the evolutionary process due to variations in the timing of selection, history of infection, and population diversity.

15. Authors claim that these findings can be further translated to the clinic. Will be good to know how? I can't see an obvious way.

16. There are couple of statements that are probably not correct. For example, in the abstract, authors use the term "coordinated adaptation". Can they explain it or remove it if this is not what they meant? Similarly, in the last paragraph of page 8, they say: "When evaluating adaptation of the specific phenotypes, we found that the colonization time had a significant impact on both growth rate and sensitivity to ciprofloxacin but did not significantly influence sensitivity to aztreonam (Figure 3C, Figure 4A and 4B), which is a reflection of the regular administration of ciprofloxacin but not aztreonam to our patients." Is there any finding supporting that?

Here is our point-by-point response to reviewers. **Text corrections are indicated by line numbers here (in red) and also in red text in our manuscript.**

Reviewer #1 (Remarks to the Author):

To study not only genetic adaptations to environmental changes but also phenotypic adaptation and to analyze how phenotype and genotype are linked to accomplish survival in host habitats is a huge challenge and promises to uncover deep and novel insight into bacterial strategies to survive during an infection process and to cause chronic infections. This study profits from a unique collection of fully genotyped *P. aeruginosa* clinical isolates that have been recovered from CF patients in a longitudinal study. These isolates were now subjected to extensive phenotyping as the starting point for the identification and characterization of complex trait adaptation strategies of *P. aeruginosa* to the lung of cystic fibrosis patients. For this purpose two models were used account for the identification of general adaptive paths that are followed by the clinical isolates and the effects of the environment on the patient-specific lineages. The impressive dataset and the original methods used in this study enabled the authors to address highly relevant questions. Their approach opens up new avenues to pursue evolutionary studies and to find new answers on mechanisms of persistence. This also promises the development of alternative strategies to fight chronic biofilm-associated infections in the future.

Specific comments:

I would have liked to see a color code in Fig 2C to distinguish early from late isolates.

Answer: We are aware that it is a complicated figure and appreciate the idea of colour coding to indicate the “age” of the individual isolates. However, we would like to highlight that the isolates are in fact ordered from early to late as shown by the lowest panel. To make this more obvious, we have made a clearer subtitle for the last panel stating “Years of colonization (by lineage)” instead of “years” and added detail to the figure legend as well to emphasize this sorting. We were concerned that a gradient of color indicating age might in fact make the figure more difficult to interpret.

Already in Fig 2 it becomes apparent that slow growth, aggregation and ciprofloxacin resistance are phenotypes that evolve in late isolates, whereas adhesion and azt resistance appear to not evolve during the course of an infection. How would the results of the AA model look like if those data were excluded?

Answer: This is an interesting question. However, based on our GAMM models, we know that only slow growth and ciprofloxacin resistance adaptation are significantly associated with colonization time when you take lineages into account, which cannot be interpreted from Figure 2. A new AA model could show new subpopulations with different extremal combinations and might inform us further about unique limitations within the evolution of slow growth versus ciprofloxacin, but the number of emergent subpopulations would be limited by only inputting 2 features. Also, this would require optimization of an entirely new model with the subset data. We believe that investigating only 2 features is better served via the GAMMs, which we have already implemented.

The authors describe consistent pan-cohort trends. Since one can group the isolates that follow distinct group-specific convergent trends, are there genetic variations that can be assigned to those groups (e.g. by applying machine learning algorithms)?

Answer: This was a driving concept behind our approach, as we had previously identified 50+ pathoadaptive genes mutated in multiple lineages of our collection. However, our collection still has quite a diverse set of mutations that appear over time in this early infection stage. We found AA patterning associated with some of the most common mutations (as shown via the ciprofloxacin resistance study), and are investigating specific mutations repeated across lineages as well as targeted operons that might associate with further patterns, but believe we need to expand the

number of patients in our study to be able to really leverage a machine-learning approach in the hunt for specific driving variants (or variant combinations). Our grouping of evolutionary trajectories is therefore not based on specific phenotypes, but rather on the mode of evolution. A limitation to using this grouping is that two lineages could be grouped by the same mode of evolution (e.g. directed evolution) but have different directions, e.g. going from A3 to A1 or from A4 to A2, which would represent the movement from different initial phenotypes to different end-point phenotypes. These different paths would then have different genes/mutations governing the adaptive trajectories. As an example, the directed evolution towards A1 could be achieved through a *gyrA* mutation whereas that would likely not lead to a trajectory going towards A2 (as seen by Figure 5A, where most isolates around A1 have a *gyrA* mutation and most isolates around A2 have a *gyrB* mutation). As the different modes of evolution (e.g. convergent vs. directed) can move towards the same archetypes, these would probably be found to have similar genes mutated. We have added additional clarifying text (lines 367-369) to explain this further.

I am not convinced by the explanation of the unexpected phenotype of a segregation of retS and retS/gacA mutants (line 339-344). The retS/gacA double mutation might drive the evolution of a phenotype that is not sufficiently described with the attributes slow growth, aggregation and resistance. It thus might be of particular interest to include virulence (cytotoxicity) data in the AA model in the future.

Answer: The original 'binary switch' theory associated with this system was formulated by Goodman et al. (2004) who included virulence impacts of the system. Our hypothesis was grounded in similar 'switch' assumptions, but we agree that the addition of other phenotypes might illustrate a trajectory shift that is not a direct reversion. At the moment, we are working on a framework for including more phenotypes and more types of data (continuous, binary and factorial) into our approach which, as suggested by the reviewer, would better illuminate the true functions of this regulatory pathway.

The "non-clustering" of the gyrB mutants (as opposed to the gyrA and gyrA/gyrB mutants) with the A1 and A6 archetypes might be explained by a lower resistance level due to the gyrB mutation. Did the authors test for the CIP MIC in their engineered PAO1 isolates?

Answer: We did test the ciprofloxacin resistance in the engineered strains but did not find a significant difference between the different mutations. We have now included this information in lines 327-330.

Reviewer #2 (Remarks to the Author):

Pseudomonas aeruginosa causes chronic airway infections in an overwhelming majority of individuals with cystic fibrosis. Previous studies have identified a myriad of adaptations (both genetic and phenotypic) that occur during the transition from initial colonization to chronic or persistent infection. This study by Bartell and colleagues examines the evolutionary trajectory of *P. aeruginosa* during adaptation to the host environment by characterizing the phenotypic and genotypic changes that occur in longitudinal isolates from 39 patients spanning a 10-year period following initial colonization. This study is impressive in scope and provides an innovative analysis approach that is biologically informative and applicable to other complex and rapidly evolving systems. This study will be influential to both the CF and *P. aeruginosa* research fields as well as the larger fields of microbial ecology, pathogenesis and evolution. Major findings include evidence for 1) a period of rapid adaptation during the first 2-3 years of colonization, and 2) simultaneous directional and diversifying trajectories. Importantly, the authors demonstrate the power of studying both phenotypic and genetic adaptations in order to fully understand the evolutionary trajectory of bacterial populations.

In general, this is a superb manuscript that could be further improved by addressing the following comments:

1. *My primary criticism is that the major findings are difficult to identify due to the overly complex presentation style. At times the manuscript is redundant and often delves too deeply into peripheral or esoteric topics of evolution. The introduction and results sections should be significantly shortened to focus on the specific findings. Speculation and interpretations should be moved to a more formal discussion section.*

Answer: We regret that the reviewer finds the manuscript overly complex. Therefore, we have done as suggested and moved several lines of speculation and interpretation to the discussion section as well as simplified some parts of the results section.

The moved or simplified lines are:

- Lines from the section “Phenotypic trends contrast with CF paradigms” regarding mucoidity (now in the discussion paragraph 2, lines 402-411)
- Lines from the first and third paragraph of the section “Initial adaptation happens within 3 years of colonization” regarding comparisons with the Lenski study of *E. coli* (now in the discussion paragraph 3, lines 411-417)
- Lines 282-286 from the section “Multi-trait analysis enables complex genotype-phenotype associations”

2. *The reliance on single isolates is problematic. While I appreciate that it is impossible to address this in a retrospective isolate collection, the authors should at least comment on how this could affect their interpretations.*

Answer: We appreciate the comment and have inserted a small section in the discussion (lines 426-436) addressing this concern by elucidating what the study setup could mean for the results.

3. *The authors reference several previous studies that document diversification as an adaptive strategy. However, there is little mention of diversity in quorum sensing phenotypes, which has been well-studied in evolving *CFP. aeruginosa* populations.*

Answer: We appreciate the importance of quorum-sensing and have investigated pathoadaptive quorum-sensing mutants in our collection via AA distribution. However, this adds another layer of multi-trait regulatory complexity to interpret and we therefore chose to focus on only clear-cut mutant-phenotype examples in this manuscript. We do include the unexpected double mutants of the *retS-gacA/S* system which integrates with QS systems via *rsmA* because the AA phenotype is so dramatic. We also hope to pursue more subtle trait relationships linked to quorum sensing regulators such as *lasR* in future work by expanding the QS-associated phenotypes we are assaying.

4. *Hypermutators are mentioned at several places in the results and supplement but they are never defined.*

Answer: We regret to have missed this definition and have now inserted a small section (lines 708-711) in Methods describing how we define hypermutators.

5. *The authors state multiple times (lines 81, 151, 170) that they use a statistical approach that accounts for patient and environmental effects (this is also an example of the redundancy in the manuscript). Do linear models fail to give the same results? Is there evidence of patient and environmental effects? What do the authors anticipate the patient and environmental factors are?*

Answer: We chose to implement GAMMs rather than a linear modelling approach because it allows both linear and nonlinear fits without forcing initial assumptions about the underlying model form and is less biased in working with data that has a high dimension to sample ratio. We further appreciate the comment regarding evidence of these patient effects, as our failure to include relevant text is a significant oversight on our part (and is now present at lines 216-219). In our

models, there is evidence of patient (and thereby environmental) effects in that the random effect term always has a significant ($p < 0.01$) impact on the predicted variable of every model. This is likely influenced by multiple patient-linked factors. We would anticipate that the difference in host response (immune reaction to the infection) would differ between patients. Furthermore, treatment intervals are different between patients as it is given on a case-by-case basis, in the sense that *P. aeruginosa* is treated when cultured. In some patients, years can elapse between culturing (in spite of the presence of the same persisting clone type) in contrast to other patients where *P. aeruginosa* is cultured on a monthly basis. Furthermore, as most of our patients are colonized by a single clone type, using patients as random effects simultaneously allows us to correct for effects of the specific clone types which might have different baseline values of e.g. growth rate or adherence.

6. The authors should comment on the complexity of using sputum as a proxy to study lung populations. CF lung disease is very heterogeneous with different regions of the infected lung having different selective pressures. Does sputum reflect all areas of the lung? Ref#12 is relevant to this point.

Answer: Our collection actually includes both sputum samples, BAL samples, and samples from sinus surgery. These are collected depending on the patient's clinical needs and are therefore not evenly distributed throughout the collection, but we can confirm that clone types are in alignment between sample types from the same patient. We have inserted text in lines 105-108 further describing our samples. We also refer the reviewer to our response to question 5 from Reviewer 3.

7. Figure 6 provides individual examples of different evolutionary trajectories. It is unclear what distinguishes the "Convergent" and "Directed" trajectories. Also, can the remaining patients be categorized based on these "overarching" modes of evolution? Is one mode more common than the others?

Answer: We have clarified the text (lines 367-369 and 378-379) such that our definitions are clearer: we consider relatively consistent linear transitions towards a single archetype to represent convergent evolution, while directed diversity retains higher diversity as the lineage moves across the archetypal space towards adapted traits. We originally inspected the AA trajectories for every patient in our collection, but many lineages are too transient/limited in colonization time to see substantial adaptation via AA. When categorizing 18 lineages with greater than 3 years' colonization and a reasonable number of isolates spanning this timeframe, we find that 8 can be described as directed, 6 as diverse, and 4 as convergent. However, we have not yet developed a quantitative method of trajectory categorization that can overcome both interpersonal bias and the diversity of our trajectories, and therefore chose not to include this assessment in the current manuscript. We hope to add more patients and extend our current lineages in the future, which will give us the means to develop effective automated categorizations of this type.

Reviewer #3 (Remarks to the Author):

The manuscript submitted to Nature Communications by Bartell et al. reports a detailed study investigating phenotypic changes in *P. aeruginosa* strains isolated from young CF patients over 10 years. One strength of the paper is the richness of the phenotypic and genotypic data. However, the main strength of the data is the elegant statistical modeling used to make the experimental findings more understandable and visible. I recommend this paper for publication after revision. Please find below my specific comments.

1. I had hard times assessing the quality of the experimental data used for the AA and GAMMs analysis. Figure 2A is confusing as it is. There are no error values for the reference strain. It is important to do a rigorous phenotyping/comparison by including the std values for the reference

PAO1 strain. Some of the assays (such as adherence) are difficult and large error bars are expected but I would personally anticipate more precise MIC measurements. It was not clear to me whether the std values here corresponded the distribution across isolates or across biological replicates for a particular strain. Probably the former. Time axis of Figure 2B is a bit confusing as well.

Answer: We have inserted text in the figure legend stating that we use an average of the phenotype in question, as well as adding the requested mean standard deviations of the phenotype. This figure is only meant to give an overview of the phenotypes present in the dataset, and because of the quantity of the data we have chosen not to include error bars as this would further complicate the figure. We instead supply data for the average as well as the standard deviations for each isolate. As for the quality of the measurements, we refer to the methods section.

2. I noticed an alternating pattern (naïve <-> evolved) in figure 2C particularly for Cipro resistance and aggregation. Have the authors checked that?

Answer: We have checked all combinations of phenotype-phenotype associations using GAMMs and did not find a significant association (positive or negative) between ciprofloxacin resistance and aggregation as indicated by the table in Figure 3D.

3. There is no specific "persistence" definition. In the field of antibiotic resistance or cancer, persistence is used in a different context.

Answer: In this study we use persistence to describe an infection that stays for an extended period of time, not referring to the distinctive 'persister' phenotype. We find it difficult to avoid using the word persistence to encompass both chronic infections and persisting lineages (with years of samples of the same clone type) that have still not been clinically diagnosed as a chronic infection (dependent on antibody levels and bacterial culture patterns at Rigshospitalet). We have altered text in the first two introduction paragraphs and added a more explicit definition at line 108-112 to hopefully convey our meaning more clearly.

4. In several places, "early" phases of evolution and "rapid" evolution are used for the changes observed in the first few years. In population dynamics, a year for bacteria is quite long. Use of these terms should be justified by number of generations.

Answer: We have included generation estimates at lines 243 + 413 (our collection) and lines 250 + 417 (prior CF estimates). We are wary of being more specific, as generation time may vary across different lineages and over time as conditions and selection pressures change within the host. We also have anchored our paper as a route to new clinical diagnostic methods, where measurements by months and years is customary versus generations.

5. Growth rate measurements are done in ASM media but some other tests are done on MH agar. I don't know whether this can cause any discrepancy. I think there must be a consistency across different phenotype measurements (i.e. same media, same temperature etc). It will be helpful to show that (for a subset of isolates) phenotypic changes such as adherence do not significantly change in different media.

Answer: Growth rate measurements were performed in both ASM and LB media and showed a significant correlation. We chose to use growth rate in ASM for the AA because it was closer to the environment found in the CF airways. Both measures were included in the GAMMs to highlight this correlation as a reference for future studies. With regards to aggregation, measurements were also taken in both LB and ASM, but only ASM generally induced the phenotype and therefore we used this measure. The reason for the use of MH agar for MIC measurements is that this is the consensus method for clinical resistance measurements and thus is necessary to compare the results with the cut-offs provided by EUCAST. We measured adherence in LB because this is the standard comparable assay, and we acknowledge that this would likely be different in ASM, to the point that the adherence might be unmeasurable in this more viscous media. We acknowledge that the type of

phenotype assays and media we chose could influence our observations. In general, the limitations of *ex-vivo* experiments must be weighed against the infeasibility of similar assays *in vivo* at this scale. When designing this study, we prioritized assays and media that would allow easy comparison to other studies as well as provide high resolution (enabling our analysis approach), and we try to avoid overinterpreting the results accordingly. We have added text (lines: 403-405) in the discussion to acknowledge this predicament of using *in vitro* measures to understand *in vivo* adaptation and evolution.

6. *Authors observe retained naïve phenotypes in late colonization and justify this observation by referring to references #3,11, and 12. In my opinion, this is not the best way to address this. One very simple possibility is new infections which has to be ruled out by showing phylogenetic distances etc.*

Answer: We have clarified this using the reviewers suggestion about phylogenetic distances. We have inserted a line describing the SNPs difference between the isolates and the nearest relative. We have also made a phylogeny of the clone type that is shared between multiple patients and in one shows late highly naïve isolates. This resulted in the adaptation of Figure 2 as well as the insertion of an explanation in lines 145-157 of the Results section “Evaluating pathogen adaptation in the early stages of infection” 3rd paragraph. The phylogenetic analysis has also been added to the methods section lines 801-805.

7. *One interesting assumption in the analysis is assuming that the infections never completely clear. Is that true? Will the GAMMs analysis examining adaptation change if new infections from other CF patients are considered possible?*

Answer: Yes, it is true that in most cases the colonization/infection never really clears. We find that roughly 75% of the patients of this collection retain the same clone type from the first isolation of *P. aeruginosa* onwards. For the second question, because we include patient as a random factor together with the time since the clone type in question was first identified in the patient, we control for new infections appearing over time with different clone types. However, as observed by the reviewer, we cannot correct for patient-to-patient transmission because we simply do not have enough information to definitively establish when or if patient-to-patient transmission has occurred. We can make assumptions as to whether a transmission has occurred, but we do not want to bias our investigation by this type of assumption. That said, if a patient receives a clone type by patient-to-patient transmission and this patient represent a monoclonal infection the fact that the clone type might be somewhat adapted already at initial infection will be corrected for by the model because of the use of patients as random effects. That is, the model will look for trends between the patients considering that some patients/clone types might have higher baseline values than others. Also, based on our previous genetic analysis in Marvig 2015, only two of our lineages are suspected to have been transmitted between 5 patients based on SNP similarity evaluation, but one of the lineages induced only transient infection (less than two years of infection that has been replaced by another clone type) in two of three patients. Therefore, only one persistent transmission event is present in our study (affecting only 2 patients). Because of the simultaneous analysis of all patients in the GAMM, this should only create a small degree of noise in the data that the model should be able to handle.

8. *In couple of places, comparisons with Lenski papers are made but these are pretty vague. For instance, "In fact, our data resembles the rate of fitnesswithin the first 5000-10000 generations". This has to be elaborated. There are too many findings in Lenski papers and it is hard to remember them all and make the comparison.*

Answer: Given this and another reviewer’s feedback, we decided the best solution was to consolidate and reduce our references to Lenski’s work to two lines of the discussion where we felt it was impactful but not overly distracting (lines: 411-417).

9. AA analysis for phenotypic changes is really interesting but maybe (not surprisingly) it is not possible to use AA as effectively for the genetic changes. Looking at number of mutations is oversimplification.

Answer: We agree with the reviewer, which is why we included the analysis with the GAMMs and mutation accumulation as it is not impressive alone. However, we thought it was not unreasonable to show that even though mutations generally accumulate over time, an isolate can be well adapted with only a few mutations. More importantly, we wanted to illustrate that hypermutators with hundreds of mutations were not the drivers of the AA archetypes as discussed at line 252-254.

10. Figure 4 is difficult to follow. The figure caption is too convoluted.

Answer: We have rearranged the figure legend and added more explanatory captions to the individual figures to improve flow.

11. Figure 4A suggests a slight growth rate decrease over time. What is the slope if one simply does a linear fit? It is most likely zero. A linear fit can be included to train our eyes.

Answer: We agree with the reviewer that a linear regression would likely give a straight line, but as this method would not incorporate patient background and the interaction with time we are hesitant to merge this method with the GAMM output. As an alternative, we have added dark gray reference lines to highlight the trend shape and timeframe of initial adaptation that we propose.

12. It would be nice to see the time periods where there is selection and there is only drift. Maybe a simple dN/dS analysis could help.

Given the structure of our data it is difficult to confidently identify periods of positive/negative selection and drift. We have added a supplemental analysis where dN/dS has been analyzed for all isolates of lineages older than three years, isolates between 0-3 years and isolates between 3-n years. We did not see any significant difference between the early-late isolate groups and thus decided to only comment on the overall dN/dS of the lineages in the paper. This information is now included in lines 267-269 in the results section and Supplemental TableS2. We also inserted information in the method section, lines: 702-706.

13. I couldn't come up with a better idea but figure 5 is organized in a very complicated way. Too many colors, shapes, and sizes.

Answer: We appreciate the difficulty with this figure and have removed the specific mutation information, aligned the colors of the lineage plots with the simplex plots and removed unnecessary jitter (shifts of co-sampled isolates along the y axis for each lineage). We hope this helps sufficiently for interpretation.

14. Authors refer to distinct routes for adaptation. In my opinion, there are no distinct adaptation mechanisms. There is still selection and neutral drift as the two main drivers. Most likely, these different classes are special snapshots of the evolutionary process due to variations in the timing of selection, history of infection, and population diversity.

Answer: We appreciate the importance of this clarification, and have altered several lines in the discussion (lines 391-392 and 438-443) as well as the last part of the results (lines: 378-379 and 384-385) discussing evolutionary modes and trajectories to reflect this important distinction and discussion of the subject.

15. Authors claim that these findings can be further translated to the clinic. Will be good to know how? I can't see an obvious way.

Answer: We regret that we have not been very clear on this point. Our main translatable conclusion is that the initial adaptation happens much faster than previously thought, so treatment should perhaps be intensified already at the first appearance of a *P. aeruginosa* from a patient sample. ■

[REDACTED]

[REDACTED] This first study suggests that we may find growth rate, adhesion, and ciprofloxacin alone sufficient phenotypes for effectively mapping trajectories and significant infection progression. This has been clarified in lines: 438-443 and 445-452.

16. *There are couple of statements that are probably not correct. For example, in the abstract, authors use the term "coordinated adaptation". Can they explain it or remove it if this is not what they meant? Similarly, in the last paragraph of page 8, they say: "When evaluating adaptation of the specific phenotypes, we found that the colonization time had a significant impact on both growth rate and sensitivity to ciprofloxacin but did not significantly influence sensitivity to aztreonam (Figure 3C, Figure 4A and 4B), which is a reflection of the regular administration of ciprofloxacin but not aztreonam to our patients." Is there any finding supporting that?*

Answer: We have rephrased the abstract to use more direct language and have also included a reference to our genotypic study where we have published the antibiotic exposure of each patient, which is primarily ciprofloxacin versus aztreonam (in comparing these two specific antibiotics). We also provide a reference for this treatment strategy as the standard employed by the Copenhagen CF clinic.

Reviewers' Comments:

Reviewer #1:

Remarks to the Author:

The authors have adequately addressed all comments. I do not have any further suggestions and recommend acceptance of the manuscript.

Reviewer #2:

Remarks to the Author:

This is a very nice study. All concerns have been addressed to my satisfaction.

Reviewer #3:

Remarks to the Author:

Dear authors and editors,

I enjoyed reading both the originally submitted and revised versions of the article. I think that this article will be well received by the microbial evolution and antibiotic resistance communities.

I see no major issues with the revised version of the article. All of the points I raised are carefully addressed and I think it should be published as it is. I would like to congratulate the authors for this exciting work.